# From Overgrowth to Complex Malformations: A Novel *EZH2* Variant Reveals the Expanding Clinical Spectrum of Weaver Syndrome

**DOI:** 10.3390/children12111487

**Published:** 2025-11-03

**Authors:** Chung-Lin Lee, Chih-Kuang Chuang, Huei-Ching Chiu, Ya-Hui Chang, Yuan-Rong Tu, Yun-Ting Lo, Jun-Yi Wu, Hsiang-Yu Lin, Shuan-Pei Lin

**Affiliations:** 1Department of Pediatrics, MacKay Memorial Hospital, Taipei 10449, Taiwan; clampcage@gmail.com (C.-L.L.); g880a01@mmh.org.tw (H.-C.C.); wish1001026@gmail.com (Y.-H.C.); 2Institute of Clinical Medicine, National Yang-Ming Chiao-Tung University, Taipei 112304, Taiwan; 3International Rare Disease Center, MacKay Memorial Hospital, Taipei 10449, Taiwan; andy11tw.e347@mmh.org.tw (Y.-T.L.); wl01723138@gmail.com (J.-Y.W.); 4Department of Medicine, Mackay Medical University, New Taipei City 25245, Taiwan; 5Department of Nursing, Mackay Junior College of Medicine, Nursing and Management, Taipei 112021, Taiwan; 6Division of Genetics and Metabolism, Department of Medical Research, MacKay Memorial Hospital, Taipei 10449, Taiwan; mmhcck@gmail.com (C.-K.C.); likemaruko@hotmail.com (Y.-R.T.); 7College of Medicine, Fu-Jen Catholic University, Taipei 242062, Taiwan; 8Department of Medical Research, China Medical University Hospital, China Medical University, Taichung 40447, Taiwan; 9Department of Infant and Child Care, National Taipei University of Nursing and Health Sciences, Taipei 108306, Taiwan

**Keywords:** Weaver syndrome, *EZH2* variant, corpus callosum dysgenesis, camptodactyly, overgrowth syndrome

## Abstract

Weaver syndrome is a rare congenital overgrowth disorder caused by pathogenic *EZH2* variants. This study reports a novel *EZH2* variant associated with atypical manifestations, including severe bilateral camptodactyly and complex brain malformations. A 4-year-old Taiwanese female exhibited classical Weaver syndrome features including macrosomia, macrocephaly, hypertelorism, and developmental delay, plus atypical findings of severe bilateral camptodactyly and complex brain malformations. Neuroimaging revealed corpus callosum dysgenesis with rostral agenesis and genu hypoplasia, bilateral frontal lobe hypoplasia, and an arachnoid cyst. The patient demonstrated global developmental delay with marked motor impairment but less severely affected speech and cognition, consistent with mild intellectual disability. Whole-exome sequencing identified a novel de novo pathogenic variant in *EZH2*: c.449T>C (p.Ile150Thr), affecting a highly conserved amino acid within the SANT domain. This case broadens the clinical spectrum of Weaver syndrome by highlighting severe camptodactyly and complex brain malformations as possible *EZH2*-related manifestations. The corpus callosum dysgenesis suggests a wider role of *EZH2* in neurodevelopment than previously recognized. The novel SANT domain variant may explain the severe phenotypic presentation. The novel *EZH2* variant c.449T>C (p.Ile150Thr) expands the molecular and phenotypic spectrum of Weaver syndrome. These findings underscore the importance of comprehensive neuroimaging and molecular genetic testing in suspected cases, particularly atypical presentations.

## 1. Introduction

Weaver syndrome is a rare congenital overgrowth disorder characterized by prenatal and postnatal macrosomia, accelerated osseous maturation, distinctive craniofacial features, and variable intellectual disability [1]. First described by Weaver et al. [2] in 1974, the condition remained genetically undefined until 2011, when two independent studies identified heterozygous pathogenic variants in the *EZH2* gene as the causative factor [3,4]. *EZH2* encodes the catalytic subunit of Polycomb Repressive Complex 2 (PRC2), a key epigenetic regulator that catalyzes trimethylation of lysine 27 on histone H3 (H3K27me3), thereby establishing repressive chromatin states critical for developmental gene regulation [5].

The estimated incidence of Weaver syndrome is approximately 1 in 15,000 births, with over 90% of cases resulting from de novo mutations, as documented in recent comprehensive studies [6,7]. The phenotypic spectrum includes characteristic craniofacial features such as a broad forehead, ocular hypertelorism, large fleshy ears, and micrognathia with a distinctive horizontal chin crease. Skeletal findings consistently include advanced bone age, camptodactyly, and deep-set nails, whereas neurological features range from mild hypotonia to moderate intellectual disability, which is present in about 80% of affected individuals [8,9,10,11].

At the molecular level, the pathogenesis is linked to partial loss of *EZH2* histone methyltransferase activity, leading to reduced H3K27 trimethylation and impaired developmental gene regulation [3]. Pathogenic variants are primarily missense mutations distributed throughout the gene, with notable clustering in the SET domain that mediates enzymatic activity [1]. Recent studies have expanded the recognized clinical spectrum to include neurological malformations such as corpus callosum dysgenesis and polymicrogyria as well as severe skeletal anomalies such as pronounced camptodactyly [12,13].

Despite advances in molecular understanding, Weaver syndrome continues to demonstrate significant phenotypic variability, and atypical presentations are increasingly reported. The identification of novel clinical features, particularly complex brain malformations and severe skeletal abnormalities, is crucial for refining the phenotypic spectrum and improving diagnostic accuracy in this rare disorder [14,15].

## 2. Case Presentation

### 2.1. Patient Demographics and Initial Presentation

The patient is a 4-year-old Taiwanese female (born 18 September 2020) who was first evaluated at 15 months of age for abnormal clinical phenotype and global developmental delay. She was born at 38 + 4 weeks’ gestation with a birth weight of 4460 g (>97th percentile), a length of 56 cm (>97th percentile), and a head circumference of 35 cm (>97th percentile), consistent with prenatal macrosomia. Delivery was by normal spontaneous vaginal delivery, with Apgar scores of 8 at 1 min and 9 at 5 min. She is the second child of nonconsanguineous parents. Her older brother, born in 2018, demonstrates normal development.

Family history revealed a paternal height of 175 cm and a maternal height of 166 cm, with a predicted target height of 164.5 cm (75–85th percentile). The patient resides in Kinmen and receives follow-up care at Taipei Veterans General Hospital. Early clinical concerns included prolonged neonatal jaundice, generalized hypotonia, and distinctive craniofacial features, which prompted genetic evaluation.

### 2.2. Clinical Features and Physical Examination

Serial anthropometric assessments confirmed persistent overgrowth (Table 1). At birth: weight 4460 g (z-score +3.2, >97th percentile), length 56 cm (z-score +3.5, >97th percentile), head circumference 35 cm (z-score +2.8, >97th percentile). At 15 months: height 92 cm (z-score +3.8, >97th percentile), weight 15 kg (z-score +3.5, >97th percentile), head circumference 48 cm (z-score +2.3, 85–97th percentile). At the most recent evaluation (4 years 10 months): height 123 cm (z-score +4.1, >97th percentile), weight 26 kg (z-score +3.7, >97th percentile), head circumference 52 cm (z-score +2.1, 85–97th percentile). Body mass index was 17.2 kg/m^2^. Z-scores were calculated using WHO Child Growth Standards, confirming severe and persistent overgrowth across all anthropometric parameters throughout the follow-up period.

Physical examination revealed dysmorphic features consistent with Weaver syndrome (Figure 1). Craniofacial abnormalities included macrocephaly, prominent squared forehead, bilateral hypertelorism with epicanthal folds, downward-slanting palpebral fissures, broad nasal bridge, small mouth, and mild micrognathia without macroglossia. A horizontal chin crease—a pathognomonic feature of Weaver syndrome—was observed. The anterior fontanelle was nearly closed, and the ears were low-set and floppy.

Cardiovascular examination revealed normal heart sounds without murmurs. Abdominal examination showed hepatomegaly and a prominent umbilical hernia. Spinal assessment indicated mild thoracolumbar kyphoscoliosis. Extremity evaluation revealed bilateral curved tibiae, equinovalgus foot deformity (more pronounced on the left), possible hallux valgus, and severe bilateral camptodactyly with marked posterior finger flexion (Figure 2). Generalized mild hypotonia was noted in all extremities.

### 2.3. Developmental Assessment

Developmental evaluation demonstrated global delay with longitudinal follow-up from 15 months to current age (4 years 10 months) revealing a disproportionate pattern across developmental domains. Assessment was based on clinical observation of developmental milestones, functional evaluations from the Early Intervention Program in Kinmen, and parental report. Formal standardized neuropsychological testing (e.g., Bayley Scales, Wechsler Intelligence Scales) was not available in this remote island setting.

Gross motor milestones were significantly and persistently delayed—sitting without assistance at 10 months (mildly delayed), limited crawling, and absence of independent standing at 15 months (markedly delayed). At 4 years, she remained unable to run or jump, requiring ongoing Early Intervention Program support in Kinmen. Serial assessments documented minimal improvement in mobility function despite consistent rehabilitation, indicating severe motor impairment as the most prominent feature of her developmental profile. Fine motor skills were similarly impaired due to severe bilateral camptodactyly, which restricted dexterity and object manipulation throughout the follow-up period.

By contrast, speech and language development, while delayed, progressed more favorably than gross motor skills, with gradual acquisition of verbal communication abilities. The patient demonstrated receptive and expressive language skills that, although below age expectations, were less severely affected than her motor capabilities. Clinical assessment suggested intellectual function within the mildly impaired range, consistent with approximately 80% of individuals with Weaver syndrome [3]. It is important to note that “less severely affected” refers to a comparison with the severe motor impairment, not to normal cognitive function. This disproportionate developmental pattern—with severe motor impairment as the predominant disability and relatively less severe cognitive- linguistic delays compared to motor function—represents a notable feature distinguishing her presentation from more globally and uniformly delayed developmental profiles. No significant behavioral problems emerged during longitudinal monitoring.

### 2.4. Neuroimaging Findings

Brain magnetic resonance imaging performed on 31 March 2021, at Taipei Veterans General Hospital revealed significant structural abnormalities extending the known neurological spectrum of Weaver syndrome. The MRI examination was performed using a 1.5T scanner and reviewed by a board-certified neuroradiologist, confirming the diagnostic findings described below.

Major findings included corpus callosum dysgenesis characterized by complete agenesis of the rostrum (the most anterior portion of the corpus callosum connecting the orbital surfaces of the frontal lobes), and hypoplasia or atrophy of the genu (the anterior curved portion) and anterior body. The remaining posterior body and splenium appeared relatively preserved. Additional abnormalities included suspected bilateral frontal lobe hypoplasia, particularly affecting the prefrontal regions, and an arachnoid cyst measuring approximately in the left quadrigeminal cistern (the subarachnoid space surrounding the superior and inferior colliculi). Bilateral middle-ear effusion was also noted.

These atypical neuroimaging findings suggest that *EZH2*-related disorders may encompass broader cerebral malformations than previously recognized. The selective involvement of anterior corpus callosum structures likely contributes to impaired interhemispheric motor coordination and may affect long-term neurodevelopmental outcomes, particularly in domains of executive function and motor planning [16,17].

### 2.5. Genetic Testing and Molecular Diagnosis

Initial chromosomal microarray analysis at Taipei Veterans General Hospital showed no pathogenic copy number variants. Whole-exome sequencing subsequently identified a novel de novo pathogenic variant in *EZH2*: c.449T>C (p.Ile150Thr) (Table 2). This variant affects a highly conserved residue within the SANT domain and was classified as pathogenic according to American College of Medical Genetics guidelines. It was absent from population databases and predicted to be deleterious by multiple computational tools.

Family segregation analysis confirmed the de novo nature of the *EZH2* variant. Sanger sequencing of both parents demonstrated wild-type sequences at the corresponding position, establishing this as a spontaneous mutation (Table 3). The patient’s father and older brother also underwent whole exome sequencing, which revealed no pathogenic variants related to the patient’ phenotype.

Additionally, whole exome sequencing identified a second variant of uncertain significance in the *ALDH18A1* gene (c.2293C>T, p.Arg765Ter), inherited from the mother. This variant has been associated with connective tissue and neurological disorders but was considered less likely to explain the patient’s primary phenotype, given the dominant inheritance pattern and the characteristic features of Weaver syndrome.

The *ALDH18A1* gene encodes Δ1-pyrroline-5-carboxylate synthase (P5CS), a rate-limiting enzyme in proline and ornithine biosynthesis [18]. Pathogenic variants in *ALDH18A1* are associated with two distinct disorders: (1) autosomal recessive cutis laxa type 3A (ARCL3A), characterized by wrinkled inelastic skin, connective tissue abnormalities including joint laxity and hernias, and variable neurological involvement [19,20]; and (2) autosomal dominant spastic paraplegia 9A (SPG9A), presenting primarily with progressive lower limb spasticity and occasionally intellectual disability [21]. 

The c.2293C>T (p.Arg765Ter) variant identified in our patient is considered an incidental finding based on three lines of evidence: (1) the variant is a nonsense mutation predicted to result in premature protein truncation and has been reported in the gnomAD database with an East Asian allele frequency of 0.0006 [22]; (2) the patient’s predominant features—including severe overgrowth (birth weight 4460 g, >97th percentile; persistent tall stature with z-scores consistently >+3.5), macrocephaly (head circumference >85th percentile throughout follow-up), characteristic craniofacial dysmorphology with pathognomonic horizontal chin crease, severe bilateral camptodactyly, and complex brain malformations including corpus callosum dysgenesis—are fully consistent with Weaver syndrome and are not recognized features of *ALDH18A1*-related disorders; and (3) no second pathogenic *ALDH18A1* allele was identified that would support a recessive inheritance pattern. Therefore, while we cannot entirely exclude a subtle modifying effect, the *ALDH18A1* variant is most likely an incidental finding that does not substantially contribute to the primary phenotype, which is comprehensively explained by the de novo *EZH2* variant.

Pedigree analysis revealed a nonconsanguineous, two-generation family, with the proband being the second child of healthy parents (Figure 3). The patient’s father, aged 35 yr, is 175 cm tall, and her mother, aged 33 yr, is 166 cm tall; both exhibit normal phenotypes. The proband has an older brother, born on April 6, 2018, who demonstrates normal growth and developmental milestones at 6 yr of age. Family segregation analysis through Sanger sequencing confirmed the de novo *EZH2* variant, with both parents carrying wild-type sequences at the corresponding genomic position. The maternal inheritance of the *ALDH18A1* variant was established through targeted sequencing, which showed the mother to be a heterozygous carrier of the c.2293C>T variant, while remaining phenotypically normal. The absence of any cutis laxa, connective tissue abnormalities, or neurological features in the heterozygous carrier mother further supports the *ALDH18A1* variant as an incidental finding rather than a causative factor. The unaffected brother was not tested for the *ALDH18A1* variant, given its uncertain clinical relevance to the proband’s phenotype. This inheritance pattern strongly supports the pathogenic role of the de novo *EZH2* variant as the primary cause of the patient’s Weaver syndrome, whereas the maternally inherited *ALDH18A1* variant represents an incidental finding of uncertain clinical significance in this case.

### 2.6. Treatment and Management

The patient’s care is coordinated through a comprehensive multidisciplinary approach tailored to her complex medical needs. Ophthalmologic management includes corrective lenses for significant refractive errors, specifically hyperopia of +9.00 diopters with astigmatism of +2.25 diopters, along with amblyopia therapy. Orthopedic consultation has been initiated to address severe bilateral camptodactyly and equinovalgus foot deformities, which may require surgical intervention to optimize functional outcomes.

Nutritional support includes vitamin D3 replacement therapy (two drops daily of vitamin D3 solution) for documented insufficiency (25(OH)-D TOTAL: 25.50 ng/mL; reference range: 30–100 ng/mL), with dosing adjusted according to serum 25-hydroxyvitamin D levels. Zinc supplementation (Zinga) was also started to correct documented deficiency (serum zinc: 608 µg/L; reference range: 700–1200 µg/L). Regular monitoring of liver function is performed due to persistent hepatomegaly observed on serial abdominal ultrasonography.

Over the 3-year follow-up period (from initial evaluation at 15 months to current age of 4 years 10 months), the patient has demonstrated variable responses to interventions. She continues to receive Early Intervention Program services in Kinmen with ongoing physical and occupational therapy focusing on enhancing gross and fine motor skills, with emphasis on adaptive strategies for severe camptodactyly. Speech therapy has been provided as needed. Motor skills have shown gradual but limited improvement, with persistent inability to run or jump at age 4 years 10 months despite consistent rehabilitation efforts. Speech and language development has progressed more favorably than motor function, with gradual acquisition of verbal communication abilities. The severe bilateral camptodactyly remains functionally limiting, particularly affecting fine motor tasks and object manipulation. Orthopedic surgical intervention is being considered to improve hand function and quality of life.

The family has received comprehensive genetic counseling regarding the de novo *EZH2* variant, inheritance patterns, and recommendations for long-term medical surveillance. This includes discussion of potential cancer screening protocols, given the reported increased risk of neuroblastoma in patients with Weaver syndrome [23].

## 3. Discussion

### 3.1. Genotype–Phenotype Correlation and Domain-Specific Effects

The novel *EZH2* variant c.449T>C (p.Ile150Thr) identified in our patient represents the first reported mutation affecting this specific residue within the SANT domain of the *EZH2* protein. The SANT (SWI3, ADA2, N-CoR, and TFIIIB) domain functions as a critical histone reader that confers sensitivity to the modification state of the histone H4 tail [24]. This domain plays an essential role in chromatin targeting and substrate recognition, complementing the catalytic function of the C-terminal SET domain.

Based on the landmark study by Tatton-Brown et al. (2013), among 48 confirmed pathogenic *EZH2* variants in Weaver syndrome, the majority cluster in the SET domain (12/48, 25%), which is the catalytic domain responsible for histone methyltransferase activity [1]. In contrast, SANT domain mutations are relatively uncommon in the reported literature, though specific quantitative data are not systematically documented. Truncating mutations are rare (4/48) and occur exclusively in the terminal exon after the SET domain, suggesting that complete loss of *EZH2* function may be embryonically lethal [3].

Recent functional studies indicate that pathogenic *EZH2* variants in Weaver syndrome result in partial loss of histone methyltransferase activity, rather than complete haploinsufficiency [5,24]. The absence of early truncating mutations in reported cases supports this mechanism, as homozygous null mutations in *EZH2* are incompatible with life in murine models [25,26]. The p.Ile150Thr substitution affects a highly conserved residue within the SANT1 domain (residues 88–169), which functions as a histone reader with specificity for the histone H4 N-terminal tail [24]. The SANT1 domain forms a narrow hydrophobic groove between α1 and α2 helices that mediates H4 binding, and disruption of this interaction may impair chromatin targeting efficiency and substrate presentation to the catalytic SET domain [24].

To provide context for our novel SANT1 domain variant, we conducted a comprehensive literature review of previously reported SANT domain mutations and compared their clinical features with those observed in our patient (Table 4). Among documented cases, p.Pro132Leu (affecting residue 132) has been reported in a 16-year-old female with Weaver syndrome who developed acute myeloid leukemia and secondary hemophagocytic lymphohistiocytosis [27]. Our variant p.Ile150Thr affects residue 150, located 18 amino acids downstream within the same SANT1 domain. Both variants affect residues within the critical H4-binding interface, suggesting potential shared mechanisms of pathogenicity through impaired chromatin recognition. Notably, both patients demonstrated severe phenotypes beyond classical Weaver syndrome features, including malignancy in the p.Pro132Leu case and severe structural malformations in our patient. However, the limited number of reported SANT domain variants precludes statistical analysis of domain-specific phenotypic correlations.

Despite extensive molecular characterization of *EZH2* variants, robust genotype–phenotype correlations remain unclear [5]. The severity of clinical features does not consistently align with the degree of histone methyltransferase activity reduction observed in vitro [5], suggesting that additional factors—such as genetic background, epigenetic regulation, and stochastic developmental variation—contribute to phenotypic diversity [28]. The severe bilateral camptodactyly and complex brain malformations in our patient extend the phenotypic spectrum of Weaver syndrome. While our patient’s severe phenotype and the malignancy development in the previously reported SANT domain case [27] may suggest a potential role for SANT domain dysfunction in disease severity, the limited number of documented SANT domain variants (n = 2 with detailed clinical data) prevents definitive conclusions regarding domain-specific genotype-phenotype correlations.

The identification of corpus callosum dysgenesis and severe skeletal abnormalities in association with a SANT domain variant highlights the importance of comprehensive clinical evaluation in all Weaver syndrome cases, regardless of mutation location. The comparison presented in Table 4 demonstrates that while both SANT domain variants share classical Weaver syndrome features, they are each associated with distinct severe complications—malignancy in one case and complex structural malformations in the other. This phenotypic heterogeneity, even within the same functional domain, underscores the complexity of *EZH2*-related pathogenesis.

Future studies incorporating functional analyses of SANT domain variants, structural modeling of protein-chromatin interactions, and systematic phenotypic characterization of larger patient cohorts will be essential to elucidate the specific contributions of different *EZH2* domains to the pathogenesis of Weaver syndrome. Particular attention should be directed toward understanding whether SANT domain mutations confer a distinct risk profile for severe developmental anomalies or malignancy compared to SET domain mutations, which would have important implications for clinical surveillance and prognostic counseling.

A limitation of this study is the absence of functional validation. Although the p.Ile150Thr variant is predicted to be pathogenic by computational tools and affects a highly conserved SANT1 residue, in vitro functional assays—including H3K27me3 quantification and histone H4-binding assays—are warranted to confirm the impact on PRC2 methyltransferase activity and chromatin recognition. Such studies would clarify the mechanistic basis underlying the severe phenotype observed in our patient.

### 3.2. Expansion of the Phenotypic Spectrum

This case expands the phenotypic spectrum of Weaver syndrome by documenting severe corpus callosum dysgenesis and extreme bilateral camptodactyly. Although neurological malformations such as polymicrogyria and ventriculomegaly have been reported with *EZH2* mutations [4,5], and corpus callosum abnormalities have been documented in some cases [12,13], systematic neuroimaging is not routinely performed in all suspected cases, leading to underreporting of these features. Our findings, together with previous reports, support the recommendation for comprehensive brain imaging as part of the diagnostic evaluation, particularly given the potential impact on neurodevelopmental outcomes and seizure risk.

The corpus callosum, the largest commissural white matter structure in the brain, contains approximately 200 million axons connecting the cerebral hemispheres [15,16]. Its development is particularly vulnerable between 11 and 20 weeks of gestation, when callosal fibers first cross the midline [16]. *EZH2* plays a crucial role in cortical neuronal migration and regulation of genes involved in establishing interhemispheric connectivity [4]. The rostral agenesis and genu hypoplasia observed in our patient likely reflect disruption of commissural neuron specification and axon guidance during this critical window.

The patient’s severe bilateral camptodactyly requiring surgical consideration also represents an atypical manifestation. Camptodactyly affects approximately 1% of the general population [29] but occurs with greater frequency and severity in Weaver syndrome [1]. The degree of bilateral involvement and functional impairment observed here highlights the potential for *EZH2* mutations to markedly disrupt digit development and underscores the need for comprehensive orthopedic management.

### 3.3. Clinical Implications and Diagnostic Considerations

Recognition of corpus callosum dysgenesis as a potential feature of Weaver syndrome has important diagnostic implications. Current diagnostic criteria emphasize overgrowth, distinctive craniofacial features, and developmental delay but may not adequately capture neurological malformations [1,3]. Our findings suggest that comprehensive brain imaging should be incorporated into the diagnostic evaluation of suspected Weaver syndrome, especially in patients with unexplained developmental delays or motor dysfunction.

The subtle facial features of Weaver syndrome, particularly in older individuals, often complicate diagnosis [3,30,31]. Advanced bone age remains a consistent diagnostic marker and should prompt *EZH2* testing when identified [3]. Intellectual disability, ranging from mild to moderate in approximately 80% of patients, is frequently the presenting concern [3].

The increased malignancy risk, particularly neuroblastoma during early childhood, remains an area of clinical uncertainty [32]. While lifetime malignancy risk has been reported in some studies, the small patient population and limited long-term follow-up data preclude evidence-based surveillance guidelines. Current consensus emphasizes clinical vigilance over routine imaging, although comprehensive monitoring protocols may be warranted given the potential for malignancy development in Weaver syndrome patients.

### 3.4. Population-Specific Considerations

As one of the few molecularly confirmed Weaver syndrome cases from an Asian population, this report contributes valuable data to understanding potential population-specific patterns. Database queries (gnomAD v4.1) [22] indicate that the identified *EZH2* variant c.449T>C is absent in East Asian populations and unreported globally, confirming its novelty. Review of the literature reveals that Weaver syndrome cases from East Asian populations remain extremely rare. As of 2025, only three molecularly confirmed cases have been reported from China [33,34], and one case from Japan was identified in the landmark 2013 study by Tatton-Brown et al. [1]. To our knowledge, the current case represents the first molecularly confirmed Weaver syndrome case reported from Taiwan, highlighting the rarity of this condition in East Asian populations.

The phenotypic features observed in our patient—including macrosomia, distinctive craniofacial characteristics, camptodactyly, and developmental delay—are consistent with those reported in predominantly European cohorts [1,4], suggesting that the core clinical spectrum remains similar across populations. However, the severe bilateral camptodactyly and complex brain malformations noted in our case may represent either variant-specific effects or phenotypic variability that warrants further investigation [5,12].

The paucity of Asian cases limits robust population-specific genotype-phenotype correlation analyses. Establishing regional genetic databases and international collaborative networks will be essential to determine whether allelic frequencies, mutation hotspots, or phenotypic presentations differ meaningfully across ethnic groups [7,8]. Such efforts would enhance diagnostic accuracy and inform population-tailored surveillance protocols for this rare but clinically significant disorder.

### 3.5. Management Considerations

Management of Weaver syndrome requires a multidisciplinary approach tailored to diverse clinical manifestations. In our patient, severe bilateral camptodactyly necessitated early orthopedic evaluation and consideration of surgical correction. Conservative measures—such as progressive splinting, physical therapy, and occupational therapy—remain first-line treatments [29,35]. Surgery is generally reserved for persistent flexion contractures >60° or when functional limitations significantly impair activities of daily living [36].

Corpus callosum dysgenesis and associated frontal lobe hypoplasia highlight the need for comprehensive neurodevelopmental assessment and early intervention services. These structural abnormalities may have long-term implications for cognitive functions, including decision-making processes and adaptive neuroplasticity [37]. The patient’s motor impairment, contrasted with relatively preserved language function, underscores the benefit of targeted therapies addressing specific developmental domains. Regular neurological follow-up is essential to monitor for seizures and guide additional interventions.

Educational support through individualized education plans is critical, given the high prevalence of intellectual disability in Weaver syndrome [3]. Adaptive strategies focusing on gross and fine motor skill development may be especially beneficial, as verbal abilities are often relatively preserved.

Comprehensive genetic counseling is essential for families affected by Weaver syndrome, addressing both immediate concerns and long-term management strategies. The de novo nature of the *EZH2* variant c.449T>C (p.Ile150Thr) in this case has important implications for recurrence risk assessment and family planning.

For parents of an affected child with a confirmed de novo *EZH2* variant, the recurrence risk in future pregnancies is low but not negligible. While the empirical recurrence risk is estimated at approximately 1–2% due to the theoretical possibility of germline mosaicism [38,39], this remains substantially lower than the 50% risk associated with inherited autosomal dominant conditions. Parents should be counseled that the vast majority (>90%) of Weaver syndrome cases result from de novo mutations [3,4], and the patient’s unaffected sibling faces no increased risk of having affected offspring unless germline mosaicism is present in a parent.

For families desiring additional children, several reproductive options merit discussion. Prenatal diagnosis via amniocentesis or chorionic villus sampling can be offered in subsequent pregnancies, with molecular testing targeting the familial *EZH2* variant [40]. Preimplantation genetic testing (PGT) represents an alternative for families pursuing in vitro fertilization, allowing identification of unaffected embryos prior to transfer [41]. The decision to pursue prenatal or preimplantation testing should incorporate family values, resources, and psychological considerations, as the phenotypic severity in Weaver syndrome—while often significant—demonstrates considerable variability and may not preclude meaningful quality of life [1,3].

Extended family counseling should address the negligible risk to siblings of the proband and clarify that parental genetic testing is unnecessary for other family members, given the de novo mutation status. However, if the affected individual eventually reproduces, each offspring will face a 50% risk of inheriting the pathogenic variant [42].

Long-term follow-up should include regular growth monitoring, developmental assessment, spinal evaluation for potential cervical spine abnormalities [43], and vigilance for malignancy development, particularly neuroblastoma during early childhood [23,32]. Specifically, surveillance should incorporate (1) comprehensive physical examination at each visit with particular attention to abdominal masses or unexplained symptoms; (2) developmental and neurological assessment every 6–12 months during early childhood; (3) abdominal ultrasound and urinary catecholamine screening if clinical suspicion of neuroblastoma arises; and (4) annual spine imaging through adolescence to monitor for cervical spine abnormalities. While evidence-based protocols specific to Weaver syndrome are lacking due to disease rarity, these recommendations align with general pediatric oncology surveillance principles for genetic tumor predisposition syndromes.

Families should be informed about patient advocacy organizations and support networks that can provide resources and connections to others navigating similar diagnoses [44]. Periodic re-evaluation of the genetic diagnosis may be warranted as additional phenotypic features emerge or as new research clarifies genotype-phenotype correlations in *EZH2*-related disorders.

### 3.6. Limitations

This case report has several limitations. First, we were unable to obtain the actual MRI images for publication, as the neuroimaging was performed at an external institution. While we have provided detailed anatomical descriptions based on the formal neuroradiology report, the absence of visual documentation limits the reader’s ability to fully appreciate the extent and pattern of brain malformations.

Second, developmental assessment was limited to clinical milestone observations, functional evaluations from early intervention services, and parental reports. Formal standardized neuropsychological testing (such as the Bayley Scales of Infant Development, Wechsler Preschool and Primary Scale of Intelligence) was not available in the remote island setting where the patient resides. While clinical assessment indicated mild intellectual disability with disproportionately severe motor impairment, the absence of quantitative developmental quotient or IQ scores limits precise characterization of the cognitive phenotype. Future follow-up with formal standardized assessments would provide more objective documentation of neurodevelopmental trajectory.

Third, long-term neurodevelopmental outcomes remain incompletely characterized given the patient’s young age (4 years 10 months at most recent evaluation). Continued follow-up will be essential to document the trajectory of cognitive, motor, and behavioral development into school age and beyond, and to assess the functional impact of corpus callosum dysgenesis on learning and adaptive skills.

## 4. Conclusions

We report a novel de novo *EZH2* variant, c.449T>C (p.Ile150Thr), in a 4-year-old Taiwanese female with Weaver syndrome. This represents the first documented mutation affecting this specific residue within the SANT domain. The case broadens the recognized phenotypic spectrum of EZH2-related overgrowth disorders by identifying severe bilateral camptodactyly and complex brain malformations, including corpus callosum dysgenesis with rostral agenesis and genu hypoplasia—features not previously emphasized in the classical Weaver syndrome phenotype. The severity of skeletal and neurological manifestations observed suggests that mutations within the SANT domain, a critical histone reader for chromatin targeting, may have particularly severe developmental consequences compared with SET domain mutations that primarily impair catalytic activity.

The identification of corpus callosum dysgenesis as a potential feature of Weaver syndrome has direct clinical implications for diagnosis and management. Comprehensive brain imaging should be incorporated into standard evaluation protocols, particularly given the potential impact on neurodevelopmental outcomes and seizure risk. Similarly, the extreme bilateral camptodactyly requiring surgical consideration highlights the importance of early orthopedic assessment and multidisciplinary intervention to optimize functional outcomes during critical developmental periods.

This case adds valuable molecular and phenotypic data to the international database of *EZH2* variants, underscoring the importance of systematic case reporting from diverse populations in advancing rare disease knowledge. As one of the few molecularly confirmed Asian cases, it also highlights potential geographic and ethnic considerations in phenotypic expression and supports the implementation of standardized surveillance protocols, including neuroblastoma screening, cervical spine monitoring, and comprehensive neurodevelopmental support.

The documentation of this novel variant and its severe phenotype emphasizes the need for functional studies to clarify the pathogenic mechanisms underlying SANT domain mutations. Future research priorities should include in vitro validation through H3K27me3 quantification and histone-binding assays, the development of targeted therapies addressing the partial loss-of-function mechanism of *EZH2* variants, the establishment of collaborative international networks for systematic natural history studies, and the implementation of precision medicine approaches based on molecular subtyping of *EZH2*-related disorders. While this single case provides important insights into phenotypic expansion and domain-specific effects, long-term follow-up and functional validation remain essential to guide clinical management and develop evidence-based treatment protocols for this rare but clinically significant disorder.

## Figures and Tables

**Figure 1 children-12-01487-f001:**
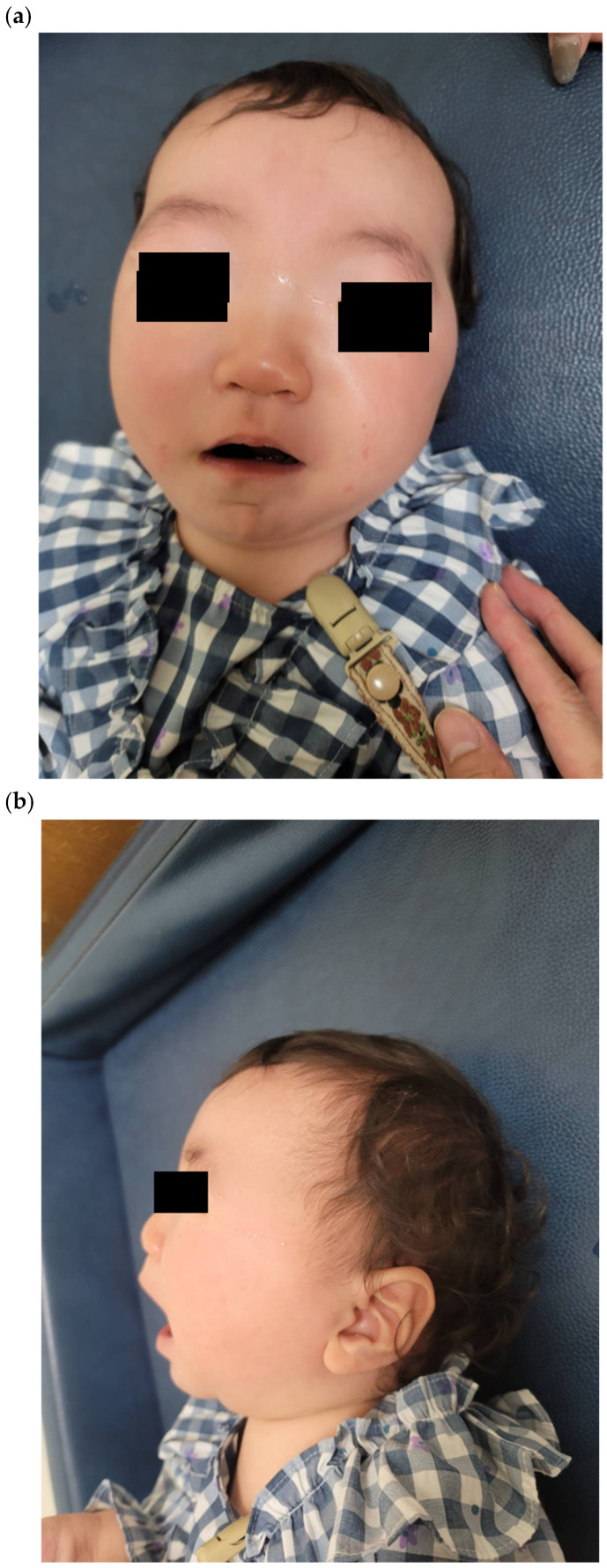
Facial dysmorphic features of the patient with Weaver syndrome. (**a**) Frontal view showing characteristic craniofacial features: macrocephaly, squared forehead, hypertelorism with epicanthal folds, downward-slanting palpebral fissures, broad nasal bridge, small mouth, and mild micrognathia. Note the horizontal chin crease, a pathognomonic feature. (**b**) Profile view showing a prominent forehead, low-set floppy ears, and facial profile consistent with Weaver syndrome. Age: 1 yr 6 months. Written informed consent was obtained from the patient’s parents for publication.

**Figure 2 children-12-01487-f002:**
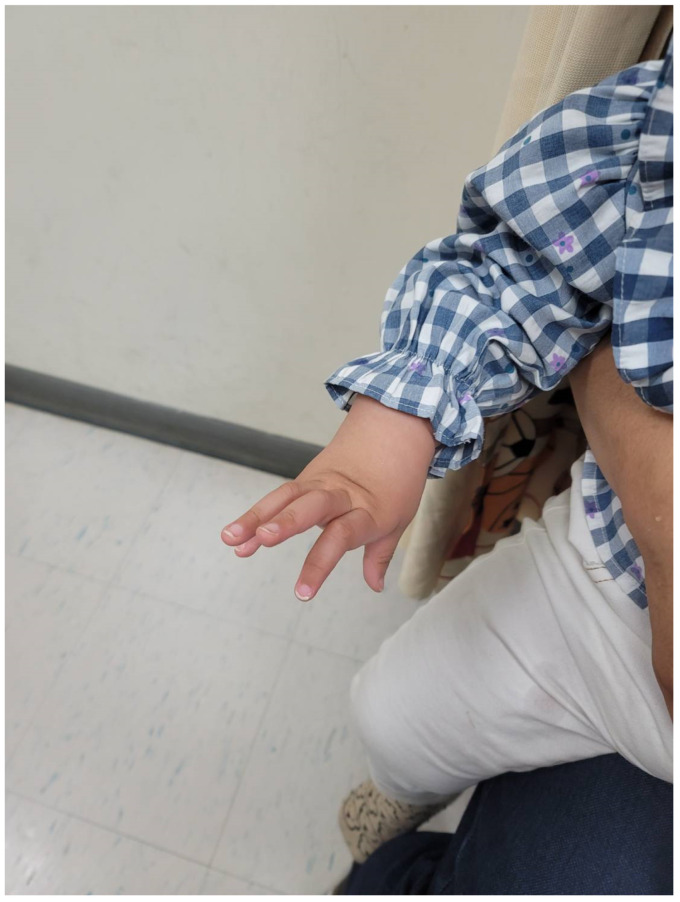
Severe bilateral camptodactyly in the patient. Clinical photograph showing marked bilateral camptodactyly with severe posterior flexion contractures of the fingers, representing an atypical and severe manifestation of skeletal abnormalities associated with Weaver syndrome. The degree of finger contracture observed here is more pronounced than typically reported in classical Weaver syndrome and results in significant functional impairment of fine motor skills. The patient was 18 months old at the time of photography. Written informed consent was obtained from the patient’s parents for publication of this image.

**Figure 3 children-12-01487-f003:**
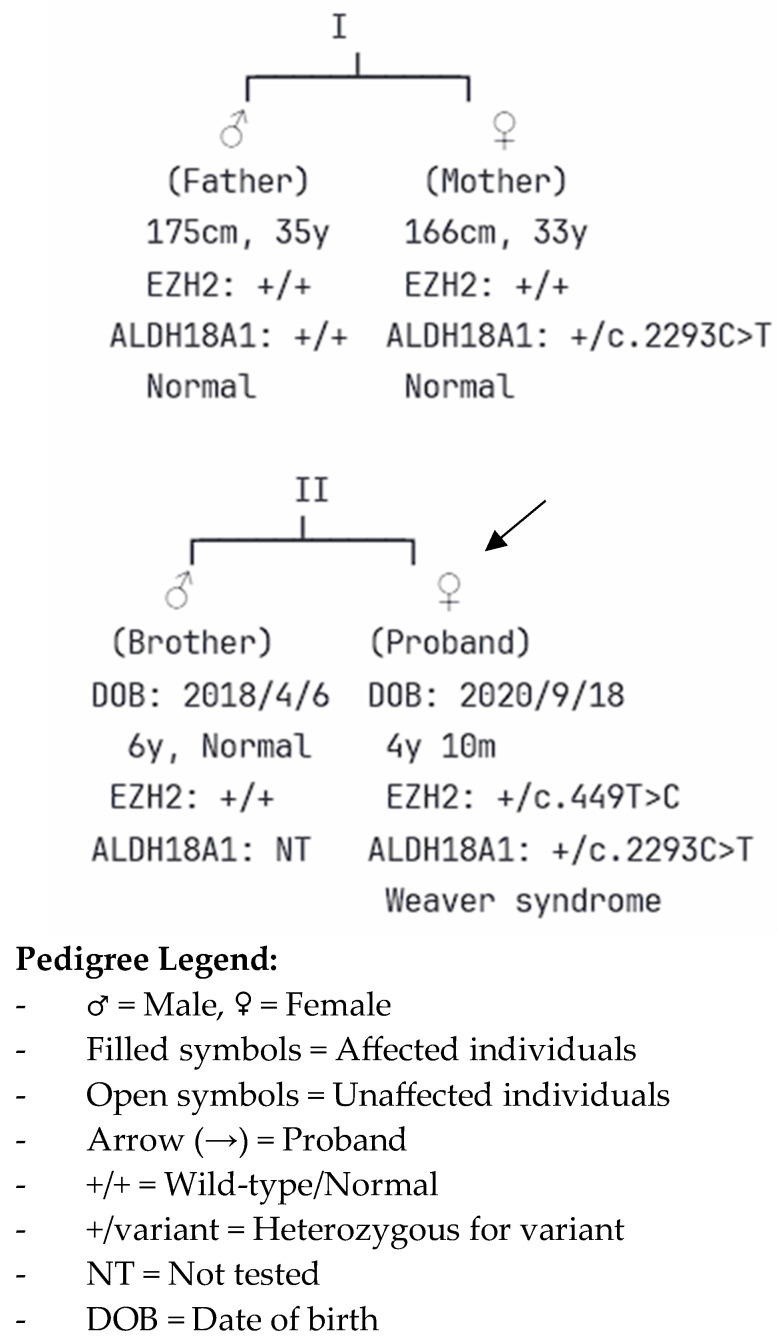
Pedigree of the Family Demonstrating Inheritance Patterns of *EZH2* and *ALDH18A1* Variants. The pedigree depicts a two-generation, nonconsanguineous family with the proband (indicated by arrow) affected by Weaver syndrome. Generation I shows unaffected parents: the father (square), 35 years old, 175 cm tall, and the mother (circle), 33 years old, 166 cm tall. Generation II includes the unaffected older brother (born in 2018) and the affected proband (born in 2020). Genetic testing results are indicated below each individual. The *EZH2* c.449T>C variant occurred de novo in the proband (heterozygous, +/c.449T>C), while both parents carried wild-type alleles (+/+). The *ALDH18A1* c.2293C>T variant was inherited from the phenotypically normal mother (heterozygous, +/c.2293C>T) to the proband, whereas the father carried wild-type alleles. The older brother was not tested for the *ALDH18A1* variant (NT = not tested). Filled symbols represent affected individuals; open symbols represent unaffected individuals. This inheritance pattern confirms the de novo pathogenic role of the *EZH2* variant as the cause of Weaver syndrome in the proband.

**Table 1 children-12-01487-t001:** Serial Anthropometric Measurements with Z-Scores.

Age	Height/Length (cm)	Height Z-Score	Weight (kg)	Weight Z-Score	Head Circumference (cm)	HC Z-Score
Birth	56	+3.5	4.46	+3.2	35	+2.8
15 months	92	+3.8	15.0	+3.5	48	+2.3
32 months	103	+3.6	19.0	+3.4	49.9	+2.2
45 months	111	+3.9	22.0	+3.6	51.1	+2.1
58 months	123	+4.1	26.0	+3.7	52.0	+2.1

Note. Z-scores calculated using WHO Child Growth Standards. All measurements consistently exceeded +2 standard deviations, confirming severe overgrowth phenotype characteristic of Weaver syndrome.

**Table 2 children-12-01487-t002:** Molecular Findings and Genetic Analysis.

Gene	Nucleotide Change	Protein Change	Inheritance	Classification	Origin	Population Frequency
*EZH2*	c.449T>C	p.Ile150Thr	Autosomal dominant	Pathogenic	De novo	Not reported
*ALDH18A1*	c.2293C>T	p.Arg765Ter	Autosomal dominant/recessive	Likely pathogenic	Maternal	0.0006 (East Asian)

Note. Molecular genetic findings were identified through whole exome sequencing. The *EZH2* variant was confirmed to be de novo through family segregation analysis, while the ALDH18A1 variant was maternally inherited. Classification was performed according to ACMG guidelines.

**Table 3 children-12-01487-t003:** Family Segregation Analysis.

Family Member	Relationship	*EZH2* c.449T>C Status	*ALDH18A1* c.2293C>T Status	Clinical Phenotype
Proband	-	Heterozygous (de novo)	Heterozygous (maternal)	Weaver syndrome with complex malformations
Father	Paternal	Wild-type	Wild-type	Normal
Mother	Maternal	Wild-type	Heterozygous	Normal
Brother	Sibling	Wild-type	Not tested	Normal development

Note. Family segregation analysis confirmed the de novo origin of the *EZH2* variant and the maternal inheritance of the *ALDH18A1* variant. Sanger sequencing was used to validate the inheritance pattern.

**Table 4 children-12-01487-t004:** Phenotypic Comparison of SANT1 Domain Variants with Typical SET Domain Presentations in Weaver Syndrome.

Feature	Current Case (Present Study)	p.Pro132Leu [27]	SET Domain Mutations [1,3,5]
Genetic Information			
Nucleotide change	c.449T>C	c.395C>T	Variable (clustered in SET)
Protein change	p.Ile150Thr	p.Pro132Leu	Variable
Domain location	SANT1 (residue 150)	SANT1 (residue 132)	SET domain (residues 612–726)
Inheritance	De novo	Germline	Predominantly de novo
Conservation	Highly conserved	Highly conserved	Highly conserved
Classical Weaver Features			
Macrosomia	+ (4460 g, >97th percentile)	+	+ (>90% of cases)
Macrocephaly	+ (>97th percentile throughout)	+	+ (>90% of cases)
Advanced bone age	Not formally assessed	+	+ (common feature)
Hypertelorism	+	+	+
Characteristic facies	+ (including horizontal chin crease)	+	+
Developmental delay	+ (global)	+	+ (~80% of cases)
Atypical/Severe Features			
Camptodactyly	Severe bilateral with marked functional impairment	Not reported in detail	Mild to moderate (common)
CNS malformations	Corpus callosum dysgenesis (rostral agenesis, genu hypoplasia), bilateral frontal lobe hypoplasia, arachnoid cyst	Not reported in detail	Rare (polymicrogyria reported in some cases)
Malignancy	None (age 4 years)	AML + secondary HLH (age 16 years)	Neuroblastoma reported (~11% risk)
Skeletal abnormalities	Severe (thoracolumbar kyphoscoliosis, curved tibiae, equinovalgus foot deformity)	Not reported in detail	Mild to moderate (variable)
Functional Implications			
Predicted effect	Impaired H4 tail binding [24]	Impaired H4 tail binding [24]	Reduced H3K27 methyltransferase activity [5]
Domain function	Histone reader (chromatin targeting)	Histone reader (chromatin targeting)	Catalytic activity (histone methylation)

Note. SANT domain variants appear to be associated with severe phenotypic presentations, though the small number of documented cases (n = 2 with detailed phenotypic data) limits definitive genotype-phenotype correlation. AML = acute myeloid leukemia; CNS = central nervous system; HLH = hemophagocytic lymphohistiocytosis; + = present.

## Data Availability

The original contributions presented in the study are included in the article. Further inquiries can be directed to the corresponding authors.

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
