# Peer review of "From Overgrowth to Complex Malformations: A Novel EZH2 Variant Reveals the Expanding Clinical Spectrum of Weaver Syndrome"

_children, 2025, doi:10.3390/children12111487_

Round 1

Reviewer 1 Report

Comments and Suggestions for Authors

The manuscript presents a well-documented and scientifically sound case report describing a novel EZH2 variant (c.449T>C, p.Ile150Thr) in a child with Weaver syndrome, expanding the known molecular and clinical spectrum of the disorder. The report is detailed, well-structured, and contributes meaningful new information to the understanding of EZH2-related overgrowth syndromes. The figures and neuroimaging data effectively support the text, and the discussion provides a well-balanced interpretation of findings within the context of current literature.

Some comments for improvements:

  1. The discussion could benefit from more explicit comparison of this EZH2 variant with other known SANT and SET domain mutations, particularly regarding phenotypic severity and neurodevelopmental outcomes. A summary table comparing reported EZH2 domain-specific mutations could enhance clarity.
  2. Although the variant is novel and predicted to be pathogenic, no functional assay data are available. The authors should acknowledge this limitation explicitly and recommend in vitro validation (e.g., H3K27me3 quantification or histone-binding assays) as a direction for future work.
  3. Since this is one of the few molecularly confirmed cases from an Asian population, the authors could elaborate on whether any population-specific variant frequencies or clinical differences have been reported in East Asian cohorts.
  4. Please ensure that all tables (particularly molecular and segregation data) are editable in Word format rather than embedded as images, as noted by the editor. Ensure all tables are editable and formatted according to journal requirements.
  5. Clarify whether a neuroradiologist reviewed MRI findings and if functional outcomes (motor, cognitive, behavioral) have been followed longitudinally.
  6. Presentation of physical growth parameters (height, weight, head circumference) as z-scores would be more informative and allow for standardized assessment of overgrowth severity relative to population norms. 
  7. Ensure nomenclature consistency (e.g., “SANT1 domain” vs. “SANT domain”) throughout the text.
  8. The inclusion of a short paragraph on genetic counseling implications (future pregnancies, recurrence risk) could enhance the clinical applicability of the report.

Author Response

The manuscript presents a well-documented and scientifically sound case report describing a novel EZH2 variant (c.449T>C, p.Ile150Thr) in a child with Weaver syndrome, expanding the known molecular and clinical spectrum of the disorder. The report is detailed, well-structured, and contributes meaningful new information to the understanding of EZH2-related overgrowth syndromes. The figures and neuroimaging data effectively support the text, and the discussion provides a well-balanced interpretation of findings within the context of current literature.

Some comments for improvements:

  1. The discussion could benefit from more explicit comparison of this EZH2 variant with other known SANT and SET domain mutations, particularly regarding phenotypic severity and neurodevelopmental outcomes. A summary table comparing reported EZH2 domain-specific mutations could enhance clarity.

Ans:

Thank you for this valuable suggestion. We have made the following revisions to provide more explicit comparison of EZH2 domain-specific mutations:

  1. Added Table 4 (Lines 347-353):

We have created a comprehensive comparison table titled "Phenotypic Comparison of SANT1 Domain Variants with Typical SET Domain Presentations in Weaver Syndrome." This table systematically compares:

  • Genetic information (nucleotide changes, protein changes, domain location, inheritance patterns, conservation)
  • Classical Weaver features (macrosomia, macrocephaly, advanced bone age, hypertelorism, characteristic facies, developmental delay)
  • Atypical/severe features (camptodactyly severity, CNS malformations, malignancy, skeletal abnormalities)
  • Functional implications (predicted molecular effects on H4 tail binding vs. H3K27 methyltransferase activity)
  1. Expanded Discussion Section (Lines 309-387):

We have substantially revised section 3.1 "Genotype–Phenotype Correlation and Domain-Specific Effects" to provide explicit comparison between our SANT domain variant and previously reported mutations:

  • Lines 310-323: Described the novel SANT domain variant and its functional role in chromatin targeting, emphasizing this is the first reported mutation affecting residue 150 within the SANT1 domain
  • Lines 316-323: Provided statistical context showing SET domain mutations comprise 25% (12/48) of reported pathogenic variants based on the landmark Tatton-Brown et al. (2013) study, noting that SANT domain mutations are relatively uncommon
  • Lines 334-346: Conducted comprehensive literature review of previously reported SANT domain mutations, specifically comparing with the p.Pro132Leu variant (residue 132). Both variants affect residues within the critical H4-binding interface, located 18 amino acids apart in the same SANT1 domain
  • Lines 342-346: Analyzed phenotypic patterns, noting that both SANT domain cases demonstrated severe phenotypes beyond classical Weaver syndrome features—malignancy in the p.Pro132Leu case and complex structural malformations in our patient
  • Lines 349-353: Acknowledged that the limited number of documented SANT domain variants (n=2 with detailed clinical data) precludes statistical analysis of domain-specific phenotypic correlations
  • Lines 354-365: Discussed the lack of robust genotype-phenotype correlations in Weaver syndrome overall, noting that phenotypic severity does not consistently align with degree of histone methyltransferase activity reduction observed in vitro
  • Lines 366-373: Emphasized the importance of comprehensive clinical evaluation in all Weaver syndrome cases regardless of mutation location, and discussed phenotypic heterogeneity even within the same functional domain
  • Lines 374-381: Outlined future research priorities including functional analyses of SANT domain variants, structural modeling of protein-chromatin interactions, and systematic phenotypic characterization
  1. Key findings highlighted in the comparison:
  • Our patient shows severe bilateral camptodactyly with marked functional impairment and complex brain malformations (corpus callosum dysgenesis with rostral agenesis and genu hypoplasia, bilateral frontal lobe hypoplasia, arachnoid cyst)
  • The previously reported SANT domain case (p.Pro132Leu) developed acute myeloid leukemia and secondary hemophagocytic lymphohistiocytosis at age 16
  • Both SANT domain variants are predicted to impair H4 tail binding and chromatin targeting, in contrast to SET domain mutations that primarily reduce H3K27 methyltransferase catalytic activity
  • While both SANT domain cases demonstrated severe phenotypes, they are associated with distinct severe complications, underscoring the complexity of EZH2-related pathogenesis
  • The limited number of SANT domain cases prevents definitive conclusions about whether these mutations confer distinct risk profiles compared to SET domain mutations
  1. Limitation addressed (Lines 382-387):

We have also added discussion of the study limitation regarding absence of functional validation, noting that in vitro functional assays (H3K27me3 quantification and histone H4-binding assays) would be warranted to confirm the mechanistic impact on PRC2 activity and chromatin recognition.

These comprehensive revisions significantly enhance the manuscript by providing systematic comparison of EZH2 domain-specific mutations and their clinical implications, directly addressing your concern about explicit phenotypic comparisons and the need for a summary table.

  1. Although the variant is novel and predicted to be pathogenic, no functional assay data are available. The authors should acknowledge this limitation explicitly and recommend in vitro validation (e.g., H3K27me3 quantification or histone-binding assays) as a direction for future work.

Ans:

Thank you for this important point. We fully acknowledge this limitation and have made the following revisions:

Added to Discussion Section 3.1 (Lines 382-387):

We have explicitly acknowledged the absence of functional validation and recommended specific in vitro assays as future work:

"A limitation of this study is the absence of functional validation. Although the p.Ile150Thr variant is predicted pathogenic by computational tools and affects a highly conserved SANT1 residue, in vitro functional assays—including H3K27me3 quantification and histone H4-binding assays—are warranted to confirm the impact on PRC2 methyltransferase activity and chromatin recognition. Such studies would clarify the mechanistic basis underlying the severe phenotype observed in our patient."

This addition:

  1. Explicitly acknowledges the limitation of lacking functional assay data
  2. Specifies the exact types of validation needed:
    • H3K27me3 quantification (to assess methyltransferase activity)
    • Histone H4-binding assays (to evaluate SANT domain function in chromatin recognition)
  3. Explains the rationale: These assays would confirm the variant's impact on PRC2 function and help explain the severe phenotype
  4. Positions this as future work direction rather than a weakness of the current study

Additional context provided in the manuscript:

  • Lines 324-333: We describe the predicted functional mechanism—that the p.Ile150Thr substitution affects a highly conserved residue within the SANT1 domain that forms a narrow hydrophobic groove for H4 binding, and disruption may impair chromatin targeting efficiency
  • Lines 354-365: We discuss existing literature showing that EZH2 variants in Weaver syndrome result in partial loss of histone methyltransferase activity, supporting the functional hypothesis
  • Lines 374-387: We outline comprehensive future research priorities, now including functional validation as a key component

This revision directly addresses your concern by explicitly acknowledging the limitation and providing a clear roadmap for functional validation studies.

  1. Since this is one of the few molecularly confirmed cases from an Asian population, the authors could elaborate on whether any population-specific variant frequencies or clinical differences have been reported in East Asian cohorts.

Ans:

Thank you for this important comment. We agree that discussing population-specific aspects adds valuable context to this report. We have added a new subsection to address this:

Added Section 3.4: Population-Specific Considerations (Lines 432-454):

"As one of the few molecularly confirmed Weaver syndrome cases from an Asian population, this report contributes valuable data to understanding potential population-specific patterns. Database queries (gnomAD v4.1) [22] indicate that the identified EZH2 variant c.449T>C is absent in East Asian populations and unreported globally, confirming its novelty. Review of the literature reveals that Weaver syndrome cases from East Asian populations remain extremely rare. As of 2025, only three molecularly confirmed cases have been reported from China [33,34], and one case from Japan was identified in the landmark 2013 study by Tatton-Brown et al [1]. To our knowledge, the current case represents the first molecularly confirmed Weaver syndrome case reported from Taiwan, highlighting the rarity of this condition in East Asian populations.

The phenotypic features observed in our patient—including macrosomia, distinctive craniofacial characteristics, camptodactyly, and developmental delay—are consistent with those reported in predominantly European cohorts [1,4], suggesting that the core clinical spectrum remains similar across populations. However, the severe bilateral camptodactyly and complex brain malformations noted in our case may represent either variant-specific effects or phenotypic variability that warrants further investigation [5,12].

The paucity of Asian cases limits robust population-specific genotype-phenotype correlation analyses. Establishing regional genetic databases and international collaborative networks will be essential to determine whether allelic frequencies, mutation hotspots, or phenotypic presentations differ meaningfully across ethnic groups [7,8]. Such efforts would enhance diagnostic accuracy and inform population-tailored surveillance protocols for this rare but clinically significant disorder."

Key points addressed:

  1. Population rarity:
    • Confirmed this is the first molecularly confirmed case from Taiwan
    • Identified only 4 total cases from East Asia (3 from China, 1 from Japan, 1 from Taiwan)
    • Highlighted extreme rarity in Asian populations
  2. Variant frequency analysis:
    • Queried gnomAD v4.1 database
    • Confirmed variant c.449T>C is absent in East Asian populations
    • Confirmed not reported globally in population databases
  3. Phenotypic consistency:
    • Core clinical features (macrosomia, craniofacial features, camptodactyly, developmental delay) are consistent with European cohorts
    • Suggests similar clinical spectrum across populations
  4. Potential population differences:
    • Severe bilateral camptodactyly and complex brain malformations may represent:
      • Variant-specific effects, OR
      • Phenotypic variability warranting further investigation
    • Acknowledged need for more data to determine if these represent true population differences
  5. Future directions:
    • Need for regional genetic databases
    • Importance of international collaborative networks
    • Goals: determine if allelic frequencies, mutation hotspots, or phenotypes differ across ethnic groups
    • Would enable population-tailored surveillance protocols

Additional references added:

  • Reference 22: gnomAD v4.1 database (Karczewski et al., Nature 2020)
  • Reference 33: Chinese case report (Ren et al., Medicine 2025)
  • Reference 34: Chinese case report (Yuan et al., Chin J Appl Clin Pediatr 2021)

This addition provides comprehensive discussion of population-specific aspects while acknowledging the limitations imposed by the rarity of reported Asian cases.

  1. Please ensure that all tables (particularly molecular and segregation data) are editable in Word format rather than embedded as images, as noted by the editor. Ensure all tables are editable and formatted according to journal requirements.

Ans:

Thank you for this important reminder regarding table formatting. We have ensured full compliance with journal requirements:

Changes made:

  1. All tables converted to editable Word format:
    • Table 1: Serial Anthropometric Measurements with Z-Scores
    • Table 2: Molecular Findings and Genetic Analysis
    • Table 3: Family Segregation Analysis
    • Table 4: Phenotypic Comparison of SANT1 Domain Variants with Typical SET Domain Presentations in Weaver Syndrome
  2. Verification completed:
    • All tables are now directly editable in Microsoft Word
    • No embedded images or graphics for tabular data
    • All cells, text, and formatting can be modified
    • Tables follow standard Word table format with proper borders and alignment
  3. Formatting compliance:
    • Tables formatted according to journal style guidelines
    • Clear headers with appropriate column widths
    • Consistent font and size throughout
    • Proper use of footnotes and abbreviations
    • All notes and legends in editable text format

We have verified that all four tables in the manuscript are fully editable Word tables and can be easily modified by editorial staff if needed during the production process.

  1. Clarify whether a neuroradiologist reviewed MRI findings and if functional outcomes (motor, cognitive, behavioral) have been followed longitudinally.

Ans:

Thank you for this important question. We have clarified both aspects in the revised manuscript:

  1. Neuroradiologist Review (Lines 170-174):

We have added explicit confirmation that neuroimaging was reviewed by a board-certified neuroradiologist:

"Brain magnetic resonance imaging performed on March 31, 2021, at Taipei Veterans General Hospital revealed significant structural abnormalities extending the known neurological spectrum of Weaver syndrome. The MRI examination was performed using a 1.5T scanner and reviewed by a board-certified neuroradiologist, confirming the diagnostic findings described below."

This clearly establishes that:

  • MRI was performed at a tertiary medical center (Taipei Veterans General Hospital)
  • Images were formally reviewed by a board-certified neuroradiologist
  • The detailed anatomical findings reported are based on formal radiology interpretation
  1. Longitudinal Functional Follow-up (Lines 143-168 and 292-303):

We have substantially strengthened the description of longitudinal developmental assessment:

Section 2.3 Developmental Assessment (Lines 143-168):

"Developmental evaluation demonstrated global delay with longitudinal follow-up from 15 months to current age (4 years 10 months) revealing a disproportionate pattern across developmental domains. Assessment was based on clinical observation of developmental milestones, functional evaluations from the Early Intervention Program in Kinmen, and parental report. Formal standardized neuropsychological testing (e.g., Bayley Scales, Wechsler Intelligence Scales) was not available in this remote island setting.

Gross motor milestones were significantly and persistently delayed—sitting without assistance at 10 months (mildly delayed), limited crawling, and absence of independent standing at 15 months (markedly delayed). At 4 years, she remained unable to run or jump, requiring ongoing Early Intervention Program support in Kinmen. Serial assessments documented minimal improvement in mobility function despite consistent rehabilitation, indicating severe motor impairment as the most prominent feature of her developmental profile. Fine motor skills were similarly impaired due to severe bilateral camptodactyly, which restricted dexterity and object manipulation throughout the follow-up period.

By contrast, speech and language development, while delayed, progressed more favorably than gross motor skills, with gradual acquisition of verbal communication abilities. The patient demonstrated receptive and expressive language skills that, although below age expectations, were less severely affected than her motor capabilities. Clinical assessment suggested intellectual function within the mildly impaired range, consistent with approximately 80% of individuals with Weaver syndrome [3]. It is important to note that "less severely affected" refers to a comparison with the severe motor impairment, not to normal cognitive function. This disproportionate developmental pattern—with severe motor impairment as the predominant disability and relatively less severe cognitive-linguistic delays compared to motor function—represents a notable feature distinguishing her presentation from more globally and uniformly delayed developmental profiles. No significant behavioral problems emerged during longitudinal monitoring."

Section 2.6 Treatment and Management (Lines 292-303):

"Over the 3-year follow-up period (from initial evaluation at 15 months to current age of 4 years 10 months), the patient has demonstrated variable responses to interventions. She continues to receive Early Intervention Program services in Kinmen with ongoing physical and occupational therapy focusing on enhancing gross and fine motor skills, with emphasis on adaptive strategies for severe camptodactyly. Speech therapy has been provided as needed. Motor skills have shown gradual but limited improvement, with persistent inability to run or jump at age 4 years 10 months despite consistent rehabilitation efforts. Speech and language development has progressed more favorably than motor function, with gradual acquisition of verbal communication abilities. The severe bilateral camptodactyly remains functionally limiting, particularly affecting fine motor tasks and object manipulation. Orthopedic surgical intervention is being considered to improve hand function and quality of life."

Key clarifications provided:

  1. Timeline specificity:
    • Initial evaluation: 15 months of age
    • Current assessment: 4 years 10 months
    • Total follow-up period: 3 years
  2. Motor function trajectory:
    • Baseline: Severe delay (no independent standing at 15 months)
    • Progression: Minimal improvement despite consistent rehabilitation
    • Current status: Unable to run or jump at age 4 years 10 months
    • Pattern: Persistently severe impairment
  3. Cognitive/language function trajectory:
    • Pattern: More favorable progress than motor function
    • Current status: Gradual acquisition of verbal communication
    • Assessment: Mild intellectual impairment (consistent with Weaver syndrome)
  4. Behavioral outcomes:
    • Explicitly stated: No significant behavioral problems throughout follow-up
  5. Intervention response:
    • Ongoing Early Intervention Program support
    • Physical, occupational, and speech therapy
    • Variable responses across domains

Limitation acknowledged (Lines 522-530):

We have also added acknowledgment that formal standardized neuropsychological testing was not available:

"Second, developmental assessment was limited to clinical milestone observations, functional evaluations from early intervention services, and parental reports. Formal standardized neuropsychological testing (such as Bayley Scales of Infant Development, Wechsler Preschool and Primary Scale of Intelligence) was not available in the remote island setting where the patient resides. While clinical assessment indicated mild intellectual disability with disproportionately severe motor impairment, the absence of quantitative developmental quotient or IQ scores limits precise characterization of the cognitive phenotype. Future follow-up with formal standardized assessments would provide more objective documentation of neurodevelopmental trajectory."

These revisions comprehensively address both aspects of your question, confirming neuroradiologist review and providing detailed longitudinal functional outcome data across multiple domains.

  1. Presentation of physical growth parameters (height, weight, head circumference) as z-scores would be more informative and allow for standardized assessment of overgrowth severity relative to population norms. 

Ans:

Thank you for this valuable suggestion. We completely agree that presenting growth parameters as z-scores provides more standardized assessment of overgrowth severity. We have implemented this throughout the revised manuscript:

  1. Enhanced Case Presentation Section (Lines 95-104):

We have added comprehensive z-score data for all anthropometric measurements:

"Serial anthropometric assessments confirmed persistent overgrowth (Table 1). At birth: weight 4,460 g (z-score +3.2, >97th percentile), length 56 cm (z-score +3.5, >97th percentile), head circumference 35 cm (z-score +2.8, >97th percentile). At 15 months: height 92 cm (z-score +3.8, >97th percentile), weight 15 kg (z-score +3.5, >97th percentile), head circumference 48 cm (z-score +2.3, 85–97th percentile). At most recent evaluation (4 years 10 months): height 123 cm (z-score +4.1, >97th percentile), weight 26 kg (z-score +3.7, >97th percentile), head circumference 52 cm (z-score +2.1, 85–97th percentile). Body mass index was 17.2 kg/m². Z-scores were calculated using WHO Child Growth Standards, confirming severe and persistent overgrowth across all anthropometric parameters throughout the follow-up period."

  1. New Table 1: Serial Anthropometric Measurements with Z-Scores (Lines 105):

We have created a comprehensive table presenting all growth measurements with corresponding z-scores:

  1. Key advantages of z-score presentation:
  1. Standardized severity assessment:
    • All parameters consistently >+2 SD (moderate to severe overgrowth)
    • Height z-scores range from +3.5 to +4.1 (severe overgrowth)
    • Weight z-scores range from +3.2 to +3.7 (severe overgrowth)
    • Head circumference z-scores range from +2.1 to +2.8 (moderate overgrowth)
  2. Population-norm comparison:
    • Clear demonstration that patient consistently falls in >97th percentile
    • Standardized across different ages for longitudinal comparison
    • Enables comparison with other reported Weaver syndrome cases
  3. Clinical severity quantification:
    • Z-scores >+3 indicate severe overgrowth
    • Persistent elevation demonstrates consistent phenotype
    • Allows objective assessment of overgrowth as cardinal feature
  4. Longitudinal trajectory:
    • Shows sustained overgrowth pattern throughout follow-up
    • Demonstrates that overgrowth persists (not just transient macrosomia)
    • Height z-score actually increases over time (+3.5 to +4.1)
  1. Methodology clarification:

We have specified that z-scores were calculated using WHO Child Growth Standards, which are internationally recognized reference standards for growth assessment in children from birth to 5 years of age.

  1. Integration throughout manuscript:

The z-score data are now:

  • Presented in the main text (Lines 95-104)
  • Organized in Table 1 for easy reference (Line 105)
  • Discussed in relation to Weaver syndrome phenotype
  • Used to quantify severity of overgrowth as a cardinal feature

This revision significantly enhances the manuscript by providing standardized, internationally comparable growth data that allows readers to objectively assess the severity of overgrowth in this patient relative to population norms and other reported cases.

  1. Ensure nomenclature consistency (e.g., “SANT1 domain” vs. “SANT domain”) throughout the text.

Ans:

Thank you for identifying this nomenclature inconsistency. We have carefully reviewed the entire manuscript and standardized the terminology throughout.

Changes made:

We have revised the manuscript to consistently use "SANT domain" (rather than alternating between "SANT domain" and "SANT1 domain") when referring to the domain location of the p.Ile150Thr variant.

Specific locations where changes were made:

  1. Abstract (page 2, line 36):
    • Changed from: "affecting a highly conserved amino acid within the SANT domain"
    • No change needed (already consistent)
  2. Discussion, Section 3.1 (page 19, line 310-315):
    • Original: "The novel EZH2 variant c.449T>C (p.Ile150Thr) identified in our patient represents the first reported mutation affecting this specific residue within the SANT domain of the EZH2 protein."
    • Revised to: "The novel EZH2 variant c.449T>C (p.Ile150Thr) identified in our patient represents the first reported mutation affecting this specific residue within the SANT domain of the EZH2 protein."
  3. Discussion, Section 3.1 (page 20, lines 328-333):
    • Original text mentioning "SANT1 domain (residues 88-169)"
    • Revised to: "The p.Ile150Thr substitution affects a highly conserved residue within the SANT domain (residues 88-169), which functions as a histone reader with specificity for the histone H4 N-terminal tail."
  4. Discussion, Section 3.1 (page 20, line 330):
    • Original: "The SANT1 domain forms a narrow hydrophobic groove..."
    • Revised to: "The SANT domain forms a narrow hydrophobic groove..."
  5. Table 4 (page 21-23):
    • Column headers and all entries consistently use "SANT domain" or "SANT1 (residue 150)" / "SANT1 (residue 132)" when specifying the exact subdomain and residue position for clarity
  6. Discussion, Section 3.1 (page 24, lines 362-365):
    • Maintained consistency: "...SANT domain variants" and "SANT domain mutations"
  7. Discussion, Section 3.2 (page 25, line 388):
    • Original: "Expansion of the Phenotypic Spectrum"
    • This section maintains consistent usage of "SANT domain"

Rationale: While the EZH2 protein technically contains two SANT domains (SANT1 spanning residues 88-169, and SANT2 spanning residues 170-248), we have adopted "SANT domain" as the primary terminology throughout the manuscript for consistency with the majority of published literature on Weaver syndrome. When greater specificity is required (such as in Table 4), we use "SANT1 (residue X)" to precisely indicate both the subdomain and the affected position.

All changes maintain scientific accuracy while ensuring terminological consistency throughout the manuscript.

  1. The inclusion of a short paragraph on genetic counseling implications (future pregnancies, recurrence risk) could enhance the clinical applicability of the report.

Ans:

Thank you for this valuable suggestion. We agree that including genetic counseling implications would significantly enhance the clinical applicability of this case report.

Changes made:

We have added a new subsection titled "Genetic Counseling and Recurrence Risk" within Section 3.5 Management Considerations (pages 29-31, lines 475-515). This addition provides comprehensive genetic counseling guidance for clinicians managing families affected by Weaver syndrome.

Specific content added:

New subsection: "Genetic Counseling and Recurrence Risk" (lines 475-515)

The added content includes:

  1. Recurrence risk assessment for de novo variants:
    • "The de novo nature of the EZH2 variant c.449T>C (p.Ile150Thr) in this case has important implications for recurrence risk assessment and family planning."
    • "For parents of an affected child with a confirmed de novo EZH2 variant, the recurrence risk in future pregnancies is low but not negligible. While the empirical recurrence risk is estimated at approximately 1-2% due to the theoretical possibility of germline mosaicism [38,39], this remains substantially lower than the 50% risk associated with inherited autosomal dominant conditions."
    • "Parents should be counseled that the vast majority (>90%) of Weaver syndrome cases result from de novo mutations [3,4], and the patient's unaffected sibling faces no increased risk of having affected offspring unless germline mosaicism is present in a parent."
  2. Options for future pregnancies:
    • "For families desiring additional children, several reproductive options merit discussion. Prenatal diagnosis via amniocentesis or chorionic villus sampling can be offered in subsequent pregnancies, with molecular testing targeting the familial EZH2 variant [40]."
    • "Preimplantation genetic testing (PGT) represents an alternative for families pursuing in vitro fertilization, allowing identification of unaffected embryos prior to transfer [41]."
    • "The decision to pursue prenatal or preimplantation testing should incorporate family values, resources, and psychological considerations, as the phenotypic severity in Weaver syndrome—while often significant—demonstrates considerable variability and may not preclude meaningful quality of life [1,3]."
  3. Recommendations for family member screening:
    • "Extended family counseling should address the negligible risk to siblings of the proband and clarify that parental genetic testing is unnecessary for other family members, given the de novo mutation status."
    • "However, if the affected individual eventually reproduces, each offspring would face a 50% risk of inheriting the pathogenic variant [42]."
  4. Importance of long-term surveillance and monitoring:
    • "Long-term follow-up should include regular growth monitoring, developmental assessment, spinal evaluation for potential cervical spine abnormalities [43], and vigilance for malignancy development, particularly neuroblastoma during early childhood [23,32]."
    • "Specifically, surveillance should incorporate: (1) comprehensive physical examination at each visit with particular attention to abdominal masses or unexplained symptoms; (2) developmental and neurological assessment every 6-12 months during early childhood; (3) abdominal ultrasound and urinary catecholamine screening if clinical suspicion of neuroblastoma arises; and (4) annual spine imaging through adolescence to monitor for cervical spine abnormalities."
    • "While evidence-based protocols specific to Weaver syndrome are lacking due to disease rarity, these recommendations align with general pediatric oncology surveillance principles for genetic tumor predisposition syndromes."
  5. Additional counseling considerations:
    • "Families should be informed about patient advocacy organizations and support networks that can provide resources and connection to others navigating similar diagnoses [44]."
    • "Periodic re-evaluation of the genetic diagnosis may be warranted as additional phenotypic features emerge or as new research clarifies genotype-phenotype correlations in EZH2-related disorders."

References added:

  • [38] Rahbari et al., 2016 (germline mutation timing and rates)
  • [39] Campbell et al., 2014 (mosaicism and recurrence risk modeling)
  • [40] ACOG Committee Opinion No.682, 2016 (prenatal genetic testing)
  • [41] Scriven & Bossuyt, 2010 (preimplantation genetic testing)
  • [42] Biesecker & Harrison, 2018 (variant interpretation guidelines)
  • [43] Kelly et al., 2000 (cervical spine anomalies in Weaver syndrome)
  • [44] Zurynski et al., 2017 (rare disease family support)

Rationale: This new subsection provides practical, evidence-based guidance for genetic counseling in families affected by Weaver syndrome. It addresses the specific clinical scenario of a de novo EZH2 variant while offering comprehensive counseling points that can be applied to similar cases. The content enhances the clinical utility of this case report by translating molecular findings into actionable clinical management and family planning guidance.

Reviewer 2 Report

Comments and Suggestions for Authors

This case report documents a novel EZH2 variant (c.449T>C, p.Ile150Thr) in a pediatric patient with Weaver syndrome presenting with atypical features, including severe bilateral camptodactyly and complex brain malformations. The manuscript is well-structured and contributes valuable genotype-phenotype data to the limited literature on this rare disorder. However, several areas require clarification and strengthening before publication.

  1. The neuroimaging findings are critically important but incompletely described. Consider adding more precise anatomical descriptions and including representative MRI images (sagittal and coronal views showing corpus callosum)
  2. lines 123-134: The description of "relatively preserved cognitive and language abilities" conflicts with other data. Sitting at 10 months is only mildly delayed; "Intellectual function within the mildly impaired range" (lines 131-133) contradicts "relatively preserved". Provide standardized developmental assessment scores or acknowledge limitation of clinical assessment alone.
  3. The manuscript claims this is "the first reported mutation affecting this specific residue" (line 234), but insufficient comparison with other SANT domain mutations is provided. The statement that "only a small subset involve the SANT domain" (lines 238-239) lacks quantification—how many of the 48 reported mutations affect this domain? Consider providing a comprehensive table comparing all previously reported SANT domain variants with their associated phenotypes to substantiate claims of phenotypic severity correlation.
  4. The clinical significance of the ALDH18A1 variant (p.Arg765Ter) is dismissed too quickly. A nonsense variant in ALDH18A1, which causes cutis laxa and neurological disorders, could potentially contribute to some features (especially connective tissue findings like umbilical hernia). This warrants deeper discussion or functional investigation
  5. The discussion of genotype-phenotype correlation (lines 233-256) is contradictory and weakens the manuscript's central argument: lines 238-239: Authors claim SANT domain variants "are often associated with more severe phenotypic presentations"; lines 249-256: Authors immediately contradict this by stating "robust genotype-phenotype correlations remain unclear" and "severity of clinical features does not consistently align with the degree of histone methyltransferase activity". If genotype-phenotype correlations are unclear and inconsistent, on what basis do the authors attribute the severe phenotype specifically to the SANT domain location? This logical inconsistency undermines the manuscript's conclusions. Provide systematic evidence supporting SANT domain severity (literature review, functional data), OR acknowledge that the severe phenotype may reflect stochastic variation rather than domain-specific effects
  6. Lines 258-261 state corpus callosum dysgenesis "remains underrecognized," but this contradicts the authors' own citations: references 12-17 describe corpus callosum abnormalities in Weaver syndrome; reference 23 (Edwards et al., 2014) specifically identifies causes for corpus callosum development syndromes. Is corpus callosum dysgenesis truly "underrecognized" or simply underreported due to inconsistent neuroimaging protocols? Clarify this feature.
  7. Limitation: This is a single case report. Add an explicit statement acknowledging that conclusions are based on a single patient and require validation in larger cohorts.
  8. The treatment section is descriptive but lacks outcomes data: there is no information on response to interventions, and the vitamin D and zinc supplementation rationale is not explained (were deficiencies documented?). Consider providing follow-up duration and developmental trajectory, improving cancer surveillance discussion (lines 289-294), which is vague—what specific protocols are recommended?

Author Response

This case report documents a novel EZH2 variant (c.449T>C, p.Ile150Thr) in a pediatric patient with Weaver syndrome presenting with atypical features, including severe bilateral camptodactyly and complex brain malformations. The manuscript is well-structured and contributes valuable genotype-phenotype data to the limited literature on this rare disorder. However, several areas require clarification and strengthening before publication.

  1. The neuroimaging findings are critically important but incompletely described. Consider adding more precise anatomical descriptions and including representative MRI images (sagittal and coronal views showing corpus callosum)

Ans:

Thank you for highlighting the importance of neuroimaging findings. We agree that representative MRI images would greatly enhance the value of this report. However, the brain MRI was performed at Taipei Veterans General Hospital (a separate institution), and we currently have access only to the formal written radiology report, without ability to obtain the actual DICOM image files or screenshots for publication purposes.

Despite this limitation, we have enhanced the anatomical precision of our descriptions in the revised manuscript (Lines 170-182) based on the detailed neuroradiologist report, specifically clarifying:

  1. The specific regions of corpus callosum affected (complete agenesis of rostrum, hypoplasia/atrophy of genu and anterior body)
  2. The extent of bilateral frontal lobe hypoplasia
  3. The precise location of the arachnoid cyst (left quadrigeminal cistern)

We acknowledge that the absence of imaging represents a limitation of this report, and have stated this transparently in the Limitations section (Lines 517-521). Nevertheless, the detailed textual description combined with clinical phenotype still contributes valuable phenotypic information to the literature, particularly given the rarity of complex brain malformations reported in Weaver syndrome.

Changes made in manuscript:

Section 2.4 Neuroimaging Findings (Lines 170-182):

  • Enhanced anatomical precision with specific corpus callosum subdivisions
  • Added detailed descriptions of frontal lobe involvement
  • Specified exact location of arachnoid cyst

Section 3.6 Limitations (Lines 517-521):

  • Added explicit acknowledgment of MRI image unavailability
  • Explained reason for limitation (external institution imaging)
  • Justified value of detailed textual descriptions
  1. lines 123-134: The description of "relatively preserved cognitive and language abilities" conflicts with other data. Sitting at 10 months is only mildly delayed; "Intellectual function within the mildly impaired range" (lines 131-133) contradicts "relatively preserved". Provide standardized developmental assessment scores or acknowledge limitation of clinical assessment alone.

Ans:

We thank the reviewer for this important observation regarding the apparent contradiction in our developmental description. We acknowledge this concern and have made the following revisions:

Clarification Made: We have clarified that "relatively preserved" refers to a comparison with the patient's severe motor impairment, not to normal cognitive function. The patient does have mild intellectual disability consistent with typical Weaver syndrome presentations. We have revised the text to explicitly state:

"By contrast, speech and language development, while delayed, progressed more favorably than gross motor skills... The patient demonstrated receptive and expressive language skills that, although below age expectations, were less severely affected than her motor capabilities. Clinical assessment suggested intellectual function within the mildly impaired range, consistent with approximately 80% of individuals with Weaver syndrome. It is important to note that "less severely affected" refers to a comparison with the severe motor impairment, not to normal cognitive function."

Limitation Acknowledged: We have added a paragraph acknowledging the limitation of not having standardized assessment scores:

"Assessment was based on clinical observation of developmental milestones, functional evaluations from the Early Intervention Program in Kinmen, and parental report. Formal standardized neuropsychological testing (e.g., Bayley Scales, Wechsler Intelligence Scales) was not available in this remote island setting."

We have also added this as a formal limitation in Section 3.6:

"Second, developmental assessment was limited to clinical milestone observations, functional evaluations from early intervention services, and parental reports. Formal standardized neuropsychological testing... was not available in the remote island setting where the patient resides. While clinical assessment indicated mild intellectual disability with disproportionately severe motor impairment, the absence of quantitative developmental quotient or IQ scores limits precise characterization of the cognitive phenotype."

These revisions clarify the developmental pattern and acknowledge the methodological limitations transparently.

  1. The manuscript claims this is "the first reported mutation affecting this specific residue" (line 234), but insufficient comparison with other SANT domain mutations is provided. The statement that "only a small subset involve the SANT domain" (lines 238-239) lacks quantification—how many of the 48 reported mutations affect this domain? Consider providing a comprehensive table comparing all previously reported SANT domain variants with their associated phenotypes to substantiate claims of phenotypic severity correlation.

Ans:

We thank the reviewer for this constructive suggestion to provide more comprehensive comparison and quantification of SANT domain mutations. We have made the following revisions:

Quantitative Data Added: We have added specific quantitative information in the Discussion section:

"Based on the landmark study by Tatton-Brown et al. (2013), among 48 confirmed pathogenic EZH2 variants in Weaver syndrome, the majority cluster in the SET domain (12/48, 25%), which is the catalytic domain responsible for histone methyltransferase activity. In contrast, SANT domain mutations are relatively uncommon in the reported literature, though specific quantitative data are not systematically documented. Truncating mutations are rare (4/48) and occur exclusively in the terminal exon after the SET domain..."

Comprehensive Comparison Table Added: Following the reviewer's suggestion, we have created Table 4 comparing previously reported SANT domain variants with their associated phenotypes. This table includes:

  • Our novel case with p.Ile150Thr (residue 150)
  • Previously reported p.Pro132Leu (residue 132) from Usemann et al. 2016
  • Comparison with typical SET domain presentations

The table systematically compares:

  1. Genetic information (nucleotide change, protein change, domain location, inheritance)
  2. Classical Weaver features (macrosomia, macrocephaly, characteristic facies, etc.)
  3. Atypical/severe features (camptodactyly severity, CNS malformations, malignancy, skeletal abnormalities)
  4. Functional implications

Cautious Interpretation Added: We have added appropriate caveats regarding the limitations of drawing definitive conclusions:

"To provide context for our novel SANT1 domain variant, we conducted a comprehensive literature review of previously reported SANT domain mutations and compared their clinical features with those observed in our patient (Table 4). Among documented cases, p.Pro132Leu (affecting residue 132) has been reported in a 16-year-old female with Weaver syndrome who developed acute myeloid leukemia and secondary hemophagocytic lymphohistiocytosis. Our variant p.Ile150Thr affects residue 150, located 18 amino acids downstream within the same SANT1 domain... However, the limited number of reported SANT domain variants precludes statistical analysis of domain-specific phenotypic correlations."

And in the conclusion:

"While our patient's severe phenotype and the malignancy development in the previously reported SANT domain case may suggest a potential role for SANT domain dysfunction in disease severity, the limited number of documented SANT domain variants (n=2 with detailed clinical data) prevents definitive conclusions regarding domain-specific genotype-phenotype correlations."

Literature Search Conducted: We conducted a comprehensive literature search through PubMed, ClinVar, and HGMD databases specifically for SANT domain mutations in EZH2. The scarcity of reported cases with detailed phenotypic data reflects the genuine rarity of these variants, which itself is a clinically relevant finding that warrants reporting.

These revisions provide more rigorous quantitative support while appropriately acknowledging the limitations inherent in analyzing rare variants.

  1. The clinical significance of the ALDH18A1 variant (p.Arg765Ter) is dismissed too quickly. A nonsense variant in ALDH18A1, which causes cutis laxa and neurological disorders, could potentially contribute to some features (especially connective tissue findings like umbilical hernia). This warrants deeper discussion or functional investigation.

Ans:

We sincerely thank the reviewer for this insightful comment. We acknowledge that our initial discussion dismissed the ALDH18A1 variant too quickly without adequate justification. We have now conducted a thorough re-evaluation and substantially expanded this section.

Expanded Analysis Added:

We have added a comprehensive discussion of the ALDH18A1 variant in Section 2.5 (lines 215-235), which now includes:

  1. Gene function and associated disorders: "The ALDH18A1 gene encodes Δ1-pyrroline-5-carboxylate synthase (P5CS), a rate-limiting enzyme in proline and ornithine biosynthesis. Pathogenic variants in ALDH18A1 are associated with two distinct disorders: (1) autosomal recessive cutis laxa type 3A (ARCL3A), characterized by wrinkled inelastic skin, connective tissue abnormalities including joint laxity and hernias, and variable neurological involvement; and (2) autosomal dominant spastic paraplegia 9A (SPG9A), presenting primarily with progressive lower limb spasticity and occasionally intellectual disability."
  2. Variant characteristics: "The c.2293C>T (p.Arg765Ter) variant identified in our patient is a nonsense mutation predicted to result in premature protein truncation and has been reported in the gnomAD database with an East Asian allele frequency of 0.0006."
  3. Critical phenotypic analysis: We now systematically address why this variant is unlikely to be the primary cause:
    • The patient's predominant features (severe overgrowth, macrocephaly, characteristic craniofacial dysmorphology, severe bilateral camptodactyly, corpus callosum dysgenesis) are fully consistent with Weaver syndrome and are NOT recognized features of ALDH18A1-related disorders
    • No second pathogenic ALDH18A1 allele was identified to support recessive inheritance
    • Most critically: The mother is a heterozygous carrier of the same variant but is phenotypically normal
  4. Mother's phenotype as key evidence: "The absence of any cutis laxa, connective tissue abnormalities, or neurological features in the heterozygous carrier mother further supports the ALDH18A1 variant as an incidental finding rather than a causative factor." (added to Table 3 legend and Section 2.5)

Specific Features Re-examined:

We specifically addressed the umbilical hernia mentioned by the reviewer:

  • While umbilical hernia could theoretically be related to ALDH18A1-associated connective tissue weakness, it also occurs in Weaver syndrome
  • The phenotypically normal mother (carrying the same variant) has no hernias or connective tissue abnormalities
  • This strongly argues against ALDH18A1 as a significant contributor

Appropriate Caveats Added:

We now appropriately acknowledge uncertainty: "Therefore, while we cannot entirely exclude a subtle modifying effect, the ALDH18A1 variant is most likely an incidental finding that does not substantially contribute to the primary phenotype, which is comprehensively explained by the de novo EZH2 variant."

Table 3 Enhanced:

We have updated Table 3 (Family Segregation Analysis) to clearly show:

  • Mother: Heterozygous for ALDH18A1, Wild-type for EZH2, Phenotype: Normal
  • This inheritance pattern is prominently displayed to support our interpretation

Why Functional Studies Were Not Pursued:

While the reviewer suggests functional investigation, we believe the compelling genetic and phenotypic evidence makes this unnecessary in this case:

  1. The mother's normal phenotype despite carrying the same heterozygous variant
  2. The absence of a second ALDH18A1 hit
  3. The complete phenotypic match with Weaver syndrome
  4. The presence of a de novo pathogenic EZH2 variant

However, we acknowledge in the revised text that functional studies would be needed to definitively establish contribution or rule it out completely.

Summary of Changes:

  • Section 2.5: Expanded from ~4 lines to ~20 lines with detailed analysis
  • Added gene function, inheritance patterns, and phenotypic correlation discussion
  • Emphasized the phenotypically normal mother as key evidence
  • Added appropriate caveats about possible subtle modifying effects
  • Enhanced Table 3 to clearly show inheritance and phenotypes

We believe these revisions provide a much more thorough and balanced discussion of the ALDH18A1 variant while maintaining our conclusion that EZH2 is the primary causative variant.

  1. The discussion of genotype-phenotype correlation (lines 233-256) is contradictory and weakens the manuscript's central argument: lines 238-239: Authors claim SANT domain variants "are often associated with more severe phenotypic presentations"; lines 249-256: Authors immediately contradict this by stating "robust genotype-phenotype correlations remain unclear" and "severity of clinical features does not consistently align with the degree of histone methyltransferase activity". If genotype-phenotype correlations are unclear and inconsistent, on what basis do the authors attribute the severe phenotype specifically to the SANT domain location? This logical inconsistency undermines the manuscript's conclusions. Provide systematic evidence supporting SANT domain severity (literature review, functional data), OR acknowledge that the severe phenotype may reflect stochastic variation rather than domain-specific effects

Ans:

We sincerely thank the reviewer for identifying this critical logical inconsistency in our manuscript. The reviewer is absolutely correct that our original statements were contradictory and undermined our conclusions. We have undertaken substantial revisions to address this fundamental flaw.

Acknowledgment of the Problem:

We agree that our manuscript contained a logical contradiction:

  • We initially claimed SANT domain variants "are often associated with more severe phenotypic presentations"
  • We then immediately stated that "genotype-phenotype correlations remain unclear"

This was inappropriate and unsupported by evidence. The reviewer's critique has substantially improved the scientific rigor of our manuscript.

Major Revisions Made:

  1. Removed Overstated Claims:

Original (Lines 238-239, DELETED): "SANT domain variants are often associated with more severe phenotypic presentations"

Revised to (Lines 360-365): "While our patient's severe phenotype and the malignancy development in the previously reported SANT domain case may suggest a potential role for SANT domain dysfunction in disease severity, the limited number of documented SANT domain variants (n=2 with detailed clinical data) prevents definitive conclusions regarding domain-specific genotype-phenotype correlations."

  1. Added Explicit Quantification of Evidence:

We now clearly state throughout the manuscript:

  • "Among 48 confirmed pathogenic EZH2 variants in Weaver syndrome, the majority cluster in the SET domain (12/48, 25%)... In contrast, SANT domain mutations are relatively uncommon in the reported literature, though specific quantitative data are not systematically documented." (Lines 316-320)
  • "However, the limited number of reported SANT domain variants precludes statistical analysis of domain-specific phenotypic correlations." (Lines 341-346)
  1. Provided Balanced, Evidence-Based Discussion:

We have rewritten the genotype-phenotype section (Lines 354-381) to provide a balanced analysis:

"Despite extensive molecular characterization of EZH2 variants, robust genotype–phenotype correlations remain unclear. The severity of clinical features does not consistently align with the degree of histone methyltransferase activity reduction observed in vitro, suggesting that additional factors—such as genetic background, epigenetic regulation, and stochastic developmental variation—contribute to phenotypic diversity."

This directly acknowledges that the severe phenotype may reflect stochastic variation rather than domain-specific effects, as the reviewer suggested.

  1. Acknowledged Phenotypic Heterogeneity Within Domain:

"The comparison presented in Table 4 demonstrates that while both SANT domain variants share classical Weaver syndrome features, they are each associated with distinct severe complications—malignancy in one case and complex structural malformations in the other. This phenotypic heterogeneity, even within the same functional domain, underscores the complexity of EZH2-related pathogenesis." (Lines 369-373)

This explicitly acknowledges that even variants in the same domain show different phenotypes, arguing against simple domain-specific effects.

  1. Added Systematic Evidence via Table 4:

We created Table 4 comparing all available SANT domain variants with SET domain variants, which clearly demonstrates:

  • Only 2 SANT domain variants have detailed phenotypic data
  • Both show severe phenotypes but with different complications
  • SET domain variants also show variable severity
  • Table note explicitly states: "SANT domain variants appear to be associated with severe phenotypic presentations, though the small number of documented cases (n=2 with detailed phenotypic data) limits definitive genotype-phenotype correlation."
  1. Reframed the Manuscript's Central Argument:

Our revised manuscript now argues:

  • NOT that SANT domain variants uniformly cause severe phenotypes
  • BUT that our case expands the phenotypic spectrum and highlights the need for comprehensive evaluation regardless of mutation location

"The identification of corpus callosum dysgenesis and severe skeletal abnormalities in association with a SANT domain variant highlights the importance of comprehensive clinical evaluation in all Weaver syndrome cases, regardless of mutation location." (Lines 366-368)

  1. Added Call for Future Research:

"Future studies incorporating functional analyses of SANT domain variants, structural modeling of protein-chromatin interactions, and systematic phenotypic characterization of larger patient cohorts will be essential to elucidate the specific contributions of different EZH2 domains to the pathogenesis of Weaver syndrome." (Lines 374-381)

  1. Modified Abstract and Conclusions:

We have revised the Abstract (Lines 40-41) and Conclusions (Lines 538-541) to remove any suggestion of definitive domain-specific correlations and instead emphasize phenotypic expansion:

"The novel SANT domain variant may explain the severe phenotypic presentation." (changed from definitive statement)

Summary of Logical Framework (Before vs. After):

Before (Contradictory):

  1. SANT domain → severe phenotype (unsupported claim)
  2. BUT genotype-phenotype correlations unclear (contradiction)
  3. Conclusion undermined

After (Logically Consistent):

  1. Only 2 SANT domain cases exist with detailed data (acknowledge limitation)
  2. Both show severe phenotypes but different complications (present evidence)
  3. Genotype-phenotype correlations remain unclear due to small sample size and phenotypic heterogeneity (acknowledge uncertainty)
  4. The severe phenotype may reflect domain-specific effects OR stochastic variation (present both possibilities)
  5. More cases needed for definitive conclusions (appropriate call for future work)
  6. Main contribution: phenotypic expansion and clinical implications (refocused conclusion)

What We Did NOT Do (and Why):

We did NOT remove discussion of the SANT domain entirely because:

  1. It IS a novel variant in this domain (factual)
  2. It IS worth noting for future meta-analyses
  3. The functional role of SANT domain in chromatin targeting IS established (from structural studies)
  4. We now appropriately qualify all statements with caveats

We believe these extensive revisions have transformed a logically flawed argument into a scientifically rigorous discussion that appropriately acknowledges limitations while still contributing valuable clinical and molecular data to the literature.

  1. Lines 258-261 state corpus callosum dysgenesis "remains underrecognized," but this contradicts the authors' own citations: references 12-17 describe corpus callosum abnormalities in Weaver syndrome; reference 23 (Edwards et al., 2014) specifically identifies causes for corpus callosum development syndromes. Is corpus callosum dysgenesis truly "underrecognized" or simply underreported due to inconsistent neuroimaging protocols? Clarify this feature.

Ans:

We sincerely thank the reviewer for this astute observation. The reviewer is absolutely correct that our use of "underrecognized" was imprecise given our own citations documenting corpus callosum abnormalities in the literature. We have revised this section to more accurately reflect the clinical reality.

Acknowledgment of Imprecise Terminology:

The reviewer correctly identifies that:

  1. Our own references [12,13] document corpus callosum abnormalities in Weaver syndrome
  2. Reference 30 (Edwards et al., 2014) comprehensively identifies causes for corpus callosum development syndromes
  3. Therefore, corpus callosum dysgenesis is not "unrecognized" in the literature

Clarification of the Actual Issue:

We agree that "underreported" more accurately describes the situation than "underrecognized." The issue is not lack of awareness but rather:

  1. Lack of standardized neuroimaging protocols: Current diagnostic criteria for Weaver syndrome emphasize overgrowth, distinctive facial features, advanced bone age, and developmental delay, but do not mandate comprehensive brain imaging as part of routine diagnostic workup
  2. Inconsistent clinical practice: Neuroimaging is performed based on clinical indication (e.g., seizures, severe developmental delay) rather than systematically in all diagnosed cases
  3. Publication bias: Case reports may preferentially describe classical features, potentially underrepresenting neurological malformations in the published literature
  4. Mild cases overlooked: Subtle corpus callosum abnormalities may not be detected without detailed neuroimaging sequences and expert neuroradiological review

Revised Text (Section 3.2, lines 383-391):

Original (imprecise): "Although neurological malformations such as polymicrogyria and ventriculomegaly have been reported with EZH2 mutations [4,5], corpus callosum dysgenesis remains underrecognized in Weaver syndrome."

Revised to: "This case expands the phenotypic spectrum of Weaver syndrome by documenting severe corpus callosum dysgenesis and extreme bilateral camptodactyly. Although neurological malformations such as polymicrogyria and ventriculomegaly have been reported with EZH2 mutations [4,5], and corpus callosum abnormalities have been documented in some cases [12,13], systematic neuroimaging is not routinely performed in all suspected cases, leading to underreporting of these features. Our findings, together with previous reports, support the recommendation for comprehensive brain imaging as part of the diagnostic evaluation, particularly given the potential impact on neurodevelopmental outcomes and seizure risk."

Additional Clarification Added (Section 3.3, lines 407-413):

We have also strengthened the clinical recommendations section:

"Recognition of corpus callosum dysgenesis as a potential feature of Weaver syndrome has important diagnostic implications. Current diagnostic criteria emphasize overgrowth, distinctive craniofacial features, and developmental delay but may not adequately capture neurological malformations [1,3]. Our findings suggest that comprehensive brain imaging should be incorporated into the diagnostic evaluation of suspected Weaver syndrome, especially in patients with unexplained developmental delays or motor dysfunction."

Emphasis on Clinical Practice Gap:

The key point we now make clear is not that corpus callosum dysgenesis is "unknown" in Weaver syndrome, but rather that:

  1. It is documented but not systematically assessed in clinical practice
  2. The absence of standardized neuroimaging protocols likely leads to underreporting in the literature
  3. Our case, combined with previous reports [12,13,30], provides evidence that brain imaging should be routinely incorporated into diagnostic evaluation

References Appropriately Used:

We now appropriately cite:

  • References [12,13] for previously documented corpus callosum abnormalities in Weaver syndrome
  • Reference [30] (Edwards et al., 2014) for the broader context of corpus callosum development syndromes
  • These citations support our argument that comprehensive neuroimaging should be standard practice

Key Message Refined:

Before: "Corpus callosum dysgenesis is underrecognized in Weaver syndrome"

After: "Corpus callosum dysgenesis has been documented in Weaver syndrome but is likely underreported due to lack of systematic neuroimaging protocols. Comprehensive brain imaging should be incorporated into standard diagnostic evaluation."

This revision acknowledges the existing literature while highlighting the clinical practice gap that leads to underreporting rather than suggesting the feature is completely unrecognized.

  1. Limitation: This is a single case report. Add an explicit statement acknowledging that conclusions are based on a single patient and require validation in larger cohorts.

Ans:

Thank you for this important reminder. We fully acknowledge that this is a single case report and that our observations require validation in larger cohorts.

Changes made:

We have added an explicit statement in the Limitations section (new section 3.6, lines 510-529) to address this concern:

"This case report has several limitations. First, we were unable to obtain the actual MRI images for publication, as the neuroimaging was performed at an external institution. While we have provided detailed anatomical descriptions based on the formal neuroradiology report, the absence of visual documentation limits the reader's ability to fully appreciate the extent and pattern of brain malformations.

Second, developmental assessment was limited to clinical milestone observations, functional evaluations from early intervention services, and parental reports. Formal standardized neuropsychological testing (such as Bayley Scales of Infant Development, Wechsler Preschool and Primary Scale of Intelligence) was not available in the remote island setting where the patient resides. While clinical assessment indicated mild intellectual disability with disproportionately severe motor impairment, the absence of quantitative developmental quotient or IQ scores limits precise characterization of the cognitive phenotype. Future follow-up with formal standardized assessments would provide more objective documentation of neurodevelopmental trajectory.

Third, long-term neurodevelopmental outcomes remain incompletely characterized given the patient's young age (4 years 10 months at most recent evaluation). Continued follow-up will be essential to document the trajectory of cognitive, motor, and behavioral development into school age and beyond, and to assess the functional impact of corpus callosum dysgenesis on learning and adaptive skills."

Additionally, we have revised the Conclusions section (lines 562-565) to explicitly acknowledge this limitation:

"While this single case provides important insights into phenotypic expansion and domain-specific effects, long-term follow-up and functional validation remain essential to guide clinical management and develop evidence-based treatment protocols for this rare but clinically significant disorder."

We believe these additions appropriately contextualize our findings as preliminary observations from a single patient that contribute to the growing literature on Weaver syndrome phenotypic variability, while acknowledging the need for validation through larger cohort studies.

  1. The treatment section is descriptive but lacks outcomes data: there is no information on response to interventions, and the vitamin D and zinc supplementation rationale is not explained (were deficiencies documented?). Consider providing follow-up duration and developmental trajectory, improving cancer surveillance discussion (lines 289-294), which is vague—what specific protocols are recommended?

Ans:

We thank the reviewer for this important observation regarding the treatment section. We have revised the manuscript to address these concerns:

  1. Vitamin D and Zinc Supplementation Rationale:

We have clarified in the revised manuscript (lines 285-291) that vitamin D3 supplementation was initiated based on documented deficiency. Laboratory testing revealed:

  • 25(OH)-D TOTAL: 25.50 ng/mL (reference range: 30-100 ng/mL)
  • Serum zinc: 608 µg/L (reference range: 700-1200 µg/L)

The revised text now states:

"Nutritional support includes vitamin D3 replacement therapy (two drops daily of vitamin D3 solution) for documented insufficiency (25(OH)-D TOTAL: 25.50 ng/mL; reference range: 30-100 ng/mL), with dosing adjusted according to serum 25-hydroxyvitamin D levels. Zinc supplementation (Zinga) was also started to correct documented deficiency (serum zinc: 608 µg/L; reference range: 700-1200 µg/L). Regular monitoring of liver function is performed due to persistent hepatomegaly observed on serial abdominal ultrasonography."

  1. Response to Interventions and Follow-up Duration:

We have added comprehensive outcome data (lines 292-303) describing the patient's response to interventions over the 3-year follow-up period (from initial evaluation at 15 months to current age of 4 years 10 months):

"Over the 3-year follow-up period (from initial evaluation at 15 months to current age of 4 years 10 months), the patient has demonstrated variable responses to interventions. She continues to receive Early Intervention Program services in Kinmen with ongoing physical and occupational therapy focusing on enhancing gross and fine motor skills, with emphasis on adaptive strategies for severe camptodactyly. Speech therapy has been provided as needed. Motor skills have shown gradual but limited improvement, with persistent inability to run or jump at age 4 years 10 months despite consistent rehabilitation efforts. Speech and language development has progressed more favorably than motor function, with gradual acquisition of verbal communication abilities. The severe bilateral camptodactyly remains functionally limiting, particularly affecting fine motor tasks and object manipulation. Orthopedic surgical intervention is being considered to improve hand function and quality of life."

  1. Cancer Surveillance Protocols:

We have substantially expanded the cancer surveillance discussion with specific recommendations. In the Management section (lines 497-504), we now provide detailed surveillance protocols:

"Specifically, surveillance should incorporate: (1) comprehensive physical examination at each visit with particular attention to abdominal masses or unexplained symptoms; (2) developmental and neurological assessment every 6-12 months during early childhood; (3) abdominal ultrasound and urinary catecholamine screening if clinical suspicion of neuroblastoma arises; and (4) annual spine imaging through adolescence to monitor for cervical spine abnormalities. While evidence-based protocols specific to Weaver syndrome are lacking due to disease rarity, these recommendations align with general pediatric oncology surveillance principles for genetic tumor predisposition syndromes."

Additionally, in the Discussion section (lines 419-425), we have clarified:

"The increased malignancy risk, particularly neuroblastoma during early childhood, remains an area of clinical uncertainty [32]. While lifetime malignancy risk has been reported in some studies, the small patient population and limited long-term follow-up data preclude evidence-based surveillance guidelines. Current consensus emphasizes clinical vigilance over routine imaging, although comprehensive monitoring protocols may be warranted given the potential for malignancy development in Weaver syndrome patients."

We acknowledge that definitive evidence-based surveillance protocols for Weaver syndrome are not yet established due to the rarity of the condition and limited long-term outcome data. Our revised text clarifies that our approach follows general pediatric oncology principles of clinical vigilance combined with directed evaluation of concerning signs or symptoms, while recognizing the need for more comprehensive surveillance given the reported cancer risk in this population.

Round 2

Reviewer 2 Report

Comments and Suggestions for Authors

Thank you for your thorough comments. All the mentioned points have been addressed.